# Maternal TDP-43 interacts with RNA Pol II and regulates zygotic genome activation

Xiaoqing Nie[1,2,6], Qianhua Xu[3,4,6], Chengpeng Xu[1,2,6], Fengling Chen[3,4,6], Qizhi Wang[1,2], Dandan Qin[1], Rui Wang[1], Zheng Gao[5], Xukun Lu[3,4], Xinai Yang[1,2], Yu Wu[1,2], Chen Gu[1,2], Wei Xie [3,4] ✉ & Lei Li [1,2] ✉

Zygotic genome activation (ZGA) is essential for early embryonic development. However, the regulation of ZGA remains elusive in mammals. Here we report that a maternal factor TDP-43, a nuclear transactive response DNA-binding protein, regulates ZGA through RNA Pol II and is essential for mouse early embryogenesis. Maternal TDP-43 translocates from the cytoplasm into the nucleus at the early two-cell stage when minor to major ZGA transition occurs. Genetic deletion of maternal TDP-43 results in mouse early embryos arrested at the two-cell stage. TDP-43 co-occupies with RNA Pol II as large foci in the nucleus and also at the promoters of ZGA genes at the late two-cell stage. Biochemical evidence indicates that TDP-43 binds Polr2a and Cyclin T1. Depletion of maternal TDP-43 caused the loss of Pol II foci and reduced Pol II binding on chromatin at major ZGA genes, accompanied by defective ZGA. Collectively, our results suggest that maternal TDP-43 is critical for mouse early embryonic development, in part through facilitating the correct RNA Pol II configuration and zygotic genome activation.

After fertilization, the mammalian embryo reprograms the terminally differentiated gametes to totipotent blastomeres with the potency to generate a whole organism. During oocyte to embryo transition, the embryo undergoes reprogramming and many key events including maternal RNA degradation and zygotic genome activation (ZGA)[1–3]. The ZGA is the first transcription event, and is essential for embryonic development. The zygotic genome is activated as two waves, namely minor and major ZGA[4,5]. Minor ZGA occurs at the end of the one-cell stage with the expression of dozens of genes[6], followed by a major ZGA bursting at the two-cell stages with the expression of thousands of genes for early embryogenesis in mice[4]. During mouse oocyte to embryo transition, the transcription machinery of RNA Pol II undergoes loading, pre-configuration and elongation for gene expression[7]. Transiently blocking minor ZGA with a transcription elongation

inhibitor DRB leads to a majority of mouse embryos arrested development at two-cell stage[5,7]. Timely release of DRB inhibition before major ZGA results in abnormal transcription with the reduced enrichment of Pol II at major ZGA and the residual of Pol II binding at one-cell Pol II target genes, accompanied by the globally reduced expression of major ZGA and ectopic expression of one-cell Pol II targets[7]. These observations suggest Pol II is tightly regulated during the transition from minor to major ZGA in mice.

Due to transcription silence before ZGA, maternal effect genes are anticipated to play dominant roles in the regulation of RNA Pol II machinery during the ZGA. Zebrafish transcription factor Pou5f1 primes at the loci of several hundred ZGA genes and is required for the expression of these genes[8,9]. However, genetic depletion of mouse maternal Pou5f1 was shown to give rise to little effect on early

[1]State Key Laboratory of Stem Cell and Reproductive Biology, Institute of Stem Cell and Regeneration, Beijing Institute of Stem Cell and Regenerative Medicine, Institute of Zoology, Chinese Academy of Sciences, Beijing, China. [2]University of Chinese Academy of Sciences, Beijing, China. [3]Center for Stem Cell Biology and Regenerative Medicine, MOE Key Laboratory of Bioinformatics, New Cornerstone Science Laboratory, School of Life Sciences, Tsinghua University, Beijing, China. [4]Tsinghua-Peking Center for Life Sciences, Beijing, China. [5]Reproductive Medicine Center of the Third Affiliated Hospital of Guangzhou Medical University, Guangzhou, China. [6]These authors contributed equally: Xiaoqing Nie, Qianhua Xu, Chengpeng Xu, Fengling Chen. ✉e-mail: xiewei121@tsinghua.edu.cn; lil@ioz.ac.cn

embryonic development[10]. Recently, several factors, such as *Dux*, *Dppa2*, *Dppa4* and *Nelfa*, were identified as the important regulators for ZGA genes in two-cell-like embryonic stem cells (ESCs)[11–17]. However, genetic disruption experiments showed that most of these factors are dispensable for both the ZGA and early embryonic development in mice[18–20]. Therefore, the regulation of Pol II machinery remains poorly understood during the mammalian oocyte to embryo transition.

TDP-43 (Transactive response DNA binding protein 43 kDa), officially named TARDBP (Tar DNA binding protein), is an evolutionarily conserved DNA/RNA binding protein[21]. TDP-43 includes several domains composed of an N-terminal nuclear localization sequence (NLS), two RNA binding domains (RRM1 and RRM2), a nuclear export signal (NES), and a C-terminal glycine-rich domain[22]. Pathogenic aggregates/inclusions of TDP-43 in the cytoplasm of nerve cells are major features of a range of neurodegenerative diseases[23]. A combination of a toxic gain in cytoplasm and a function loss in nucleus of TDP-43 is believed to be the predominant reason for neuronal death in patients suffering from these diseases[24]. Since its aggregates have been shown to be related to neurodegenerative diseases, great efforts have been made to explore the roles of TDP-43 in different cells and animals[25,26].

Mouse *Tdp43/TDP-43/Tardbp* universally expresses and encodes a protein sharing high similarity with its human homolog[21,27]. Specific deletion of TDP-43 with *Stra8-Cre* mice results in a severe loss of male germ cells and male infertile[28]. The intercrosses of heterozygous (*Tdp43+/-*) mice could not deliver homozygous (*Tdp43-/-*) offspring and *Tdp43-/-* embryos die between embryonic day 3.5 (E3.5) and E8.5[27,29,30]. TDP-43 null blastocysts were unable to generate mouse embryonic stem cells (ESCs) due to proliferation defects of inner cell mass[27,30]. Thus, zygotic TDP-43 is essential for mouse early embryonic development. Although many mouse models have been developed to investigate TDP-43, the current understanding of its functions and mechanisms remains limited. Whether this multifunctional protein is involved in mouse oocyte to embryo transition remains unknown.

In this work, we find that TDP-43 translocates from the cytoplasm into the nucleus at the mouse early two-cell stage, a simultaneous occurrence with mouse major ZGA[31]. Depletion of TDP-43 in mouse oocytes shows that maternal TDP-43 is required for the ZGA and early embryogenesis. Detailed analyses show that TDP-43 interacts with RNA Pol II and Cyclin T1 of the P-TEFb complex, and is required for proper Pol II configuration and the expression of ZGA genes during mouse oocyte to embryo transition.

## Results

### The translocation of TDP-43 at mouse early 2-cell stage
We first investigated the expression of *Tdp43* mRNA in mouse oocytes and preimplantation embryos by analyzing the previous data of RNA-seq and Ribo-seq[32]. The results showed that the levels of *Tdp43* expression were dynamic in mouse oocytes and early embryos (Supplementary Fig. 1a). This dynamic pattern of TDP-43 expression in mouse oocytes and early embryos was confirmed by Western blotting, showing that TDP-43 remained relatively high and stable in mouse oocytes and 1-cell embryos, dramatically decreased in 2-cell and morula stage embryos, while strongly increased in blastocyst stage embryos (Supplementary Fig. 1b).

Immunofluorescent staining showed that TDP-43 protein was primarily in the nucleus throughout the oocyte growing stages, but in the cytoplasm of MII oocytes and 1-cell embryos, then reappeared in the nucleus of 2-cell embryos (Fig. 1a, Supplementary Fig. 1c). Notably, TDP-43 was still in the cytoplasm of very early 2-cell embryos (pHCG 32 h), started to appear in the nucleus of early 2-cell embryos (pHCG 33–34 h), and largely accumulated in the nucleus of following 2-cell embryos (pHCG 35-47 h) (Fig. 1b and Supplementary Fig. 1d). The timing of TDP-43 entering the nucleus is well correlated with the

minor-to-major ZGA transition in mouse two-cell embryo[7,31], raising a possibility that TDP-43 may play a role in this process.

To investigate the role of maternal TDP-43, we first mated *Tdp43flox/flox* mice with *Zp3-Cre* mice to obtain *Tdp43+/flox;Zp3-Cre* mice[33,34]. Then, we intercrossed *Tdp43+/flox;Zp3-Cre* mice to obtain *Tdp43flox/flox;Zp3-Cre* females in which TDP-43 was supposed to be disrupted in its oocytes, thus as *Tdp43oKO* (oKO, oocyte-specific knockout) mice (Materials and Methods). *Tdp43oKO* mice grew normally to their adulthood. The oocytes from *Tdp43oKO* females underwent efficient elimination of TDP-43 at both the mRNA and protein levels (Supplementary Fig. 1e–g). Compared with the controls (*Tdp43flox/flox* females) that normally produced litters (8.8 ± 2.1 pups/litter), *Tdp43oKO* females did not deliver any pups after mating with fertile wild type (WT) males for over 3 months (Supplementary Fig. 1h). Thus, these results demonstrate that maternal TDP-43 is essential for the fertility of female mice.

### Maternal TDP-43 is critical for mouse embryogenesis beyond 2-cell stage
Histological analysis showed that the follicles of different stages and corpus luteum were present in the ovaries from 8-week-old *Tdp43oKO* and control females (Supplementary Fig. 2a). The ratio of NSN (non-surrounded nucleolus) and SN (surrounded nucleolus) was comparable for the GV (germinal vesicle) oocytes from control and *Tdp43oKO* females (Supplementary Fig. 2b). Furthermore, the rate of germinal vesicle breakdown and first polar body extrusion were also not affected in these mutants (Supplementary Fig. 2c, d). Moreover, super-ovulation experiments show that the morphology and number of MII oocytes were almost normal (Supplementary Fig. 2e, f). Altogether, these results suggest that depletion of maternal TDP-43 barely affects mouse ovarian development, oocyte maturation and ovulation. Unless specially mentioned, control and *Tdp43oKO* oocytes/embryos were isolated from *Tdp43flox/flox* and *Tdp43oKO* females, respectively.

Next, we collected the control and *Tdp43oKO* embryos at embryonic day 0.5 (E0.5), E1.5 and E2.5. We isolated a similar number of 1-cell embryos with pronucleus at E0.5 (Fig. 1c, d), suggesting maternal TDP-43 is not essential for fertilization. At E1.5, although not statistically significant difference compared to the control, we obtained a reduced number of *Tdp43oKO* 2-cell embryos, including many asymmetrical embryos (Fig. 1c, d). At E2.5 when most control embryos developed into the morula stage, the majority of *Tdp43oKO* embryos were still arrested at the 2-cell stage or fragmentation (Fig. 1c, d). Taken together, these results suggest that maternal TDP-43 is critical for mouse early embryogenesis beyond the 2-cell stage.

### Maternal TDP-43 is required for the expression of ZGA genes
We applied RNA-seq to analyze gene expression for control and *Tdp43oKO* FGO (full-grown oocytes), MII, 1 C (1-cell embryos), E2C and L2C (early and late 2-cell embryos). Our RNA-seq data were well correlated between the replicates ($R > 0.95$, Supplementary Fig. 3a). Principal component analysis (PCA) suggested that the expression levels of genome-wide mRNAs were similar between the control and *Tdp43oKO* of FGOs and MII, 1 C and E2C (Fig. 2a), with few differently expressed genes (DEGs) (Supplementary Fig. 3b–e), and the majority of DEGs did not originate from the previous developmental stage (Supplementary Fig. 3f–h). These results indicate a mild, if any, defect in the *Tdp43oKO* oocytes and early embryos before the early 2-cell stage.

By contrast, substantial differences in gene expression appeared at the late 2-cell stage between control and *Tdp43oKO* embryos as indicated by the PCA analysis (Fig. 2a). Compared with the control, we identified 1402 up-regulated genes and 1319 down-regulated genes in *Tdp43oKO* L2C embryos (adjust *p*-value, 0.05; Fold change, 2) (Fig. 2b). Maternal RNA degradation and ZGA are two major events during mammalian maternal to zygotic transition[2]. The global maternal RNA degradation was not evidently affected in TDP-43 mutants compared

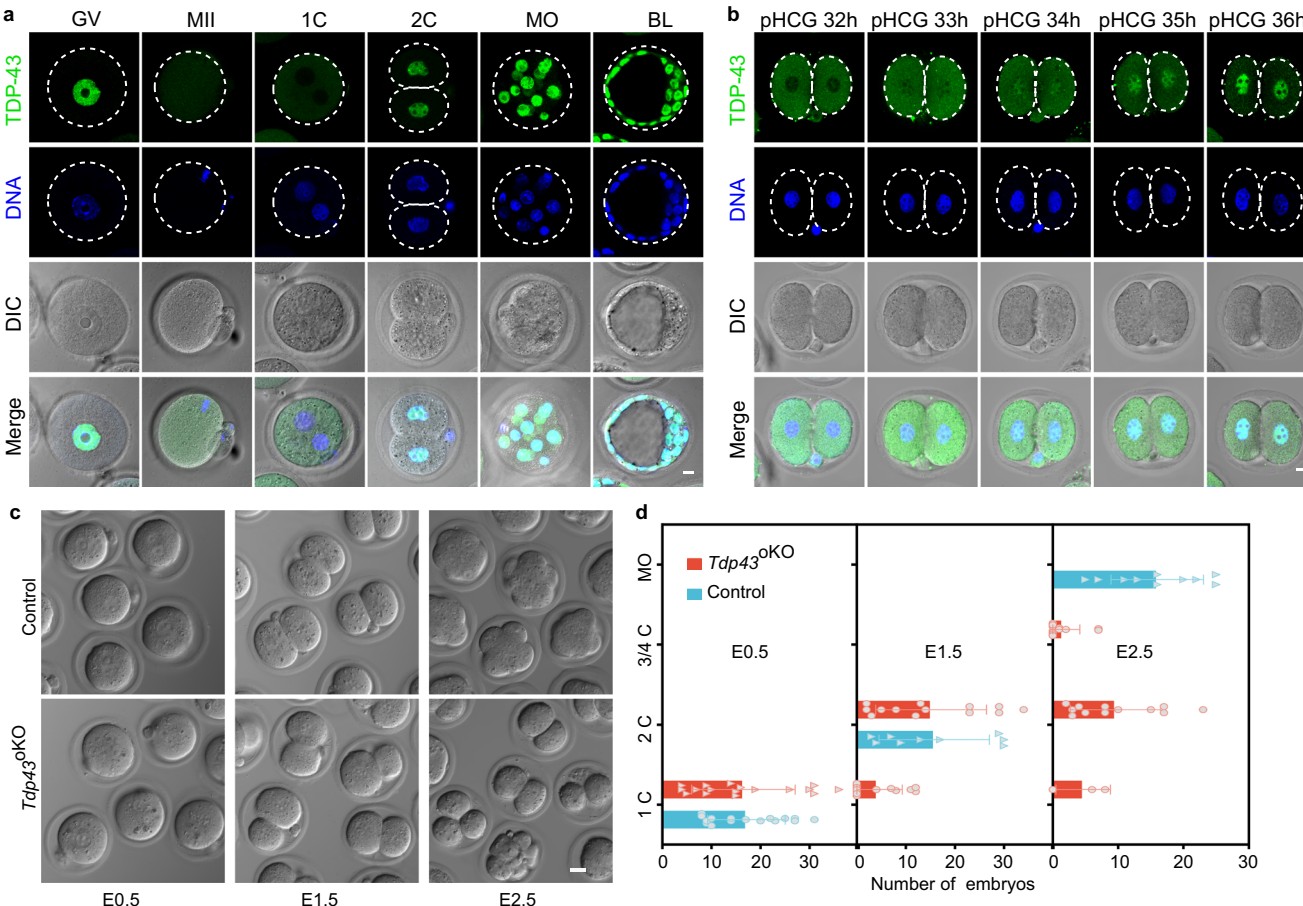

**Fig. 1 | TDP-43 translocates from the cytoplasm into the nucleus at the mouse early 2-cell stage and is essential for early embryogenesis. a** Immunostaining of TDP-43 in mouse oocytes and early embryos. GV germinal vesicle oocyte, MII MII oocyte; 1 C 1-cell embryo, 2 C 2-cell embryo, MO morula, BL blastocyst. Scale bar, 10 μm. **b** Immunostaining of TDP-43 in the 2-cell embryo at different time points. pHCG post HCG injection, h hours. Scale bar, 10 μm. **c** DIC images of different stage embryos isolated at embryonic day 0.5 (E0.5), E1.5 and E2.5 from the oviducts of control (*Tdp43^flox/flox*) and *Tdp43*^oKO (*Tdp43 ^flox/flox*;*Zp3-Cre*) females mated with normal fertile males. Scale bar, 20 μm. **d** The average numbers of different stage embryos that were flushed at E0.5, E1.5 and E2.5 from the oviducts of control (*n* = 16, *n* = 9, *n* = 10) and *Tdp43*^oKO (*n* = 17, *n* = 13, *n* = 12) females, the dots represent numbers of the embryos from each female. Error bars, SD (standard deviation).

with the control (Fig. 2c, Supplementary Fig. 4a). By contrast, many minor ZGA genes which should be decreased in L2C showed the up-regulated expression, and substantial major ZGA genes showed the down-regulated expression when compared with their controls (Fig. 2c). About 56.9% of minor ZGA genes (37/65) were significantly up-regulated and ~23.7% of major ZGA genes (263/1107) were significantly down-regulated in the *Tdp43*^oKO L2C embryos. Altogether, these results suggest that the depletion of maternal TDP-43 leads to an aberrant transition from minor ZGA to major ZGA.

To further confirm the aberrant ZGA is directly caused by TDP-43 depletion in *Tdp43*^oKO L2C embryos, we knocked down TDP-43 protein in mouse 1-cell embryos using Trim-away with the specific antibody against TDP-43 and cultured these embryos to the L2C stage (Materials and Methods)[35]. TDP-43 was efficiently knock-down in the embryos that were injected with *Trim21* mRNA and the antibody against TDP-43 (hereafter KD embryos), but not in the Inj-Contr (injection control with *Trim21* mRNA and IgG), probably because of TDP-43 localization in the cytoplasm until early 2-cell stage (Fig. 1b, Supplementary Fig. 4b). We collected twenty KD and Inj-Contrl embryos for Smart-seq2 analysis. Our results showed that these RNA-seq data were well correlated between the replicates (Supplementary Fig. 4c, d). Compared with Inj-Contrl, 727 and 864 genes were identified to be down-regulated and up-regulated in KD L2C embryos by RNA-seq analysis (adjust *p*-value, 0.05; Fold change, 2) (Supplementary Fig. 4e), respectively, including plenty of ZGA genes (Supplementary Fig. 4f, g). Among these genes,

~52.8% (384/727) also showed down-regulation in *Tdp43*^oKO embryos, and ~30.7% (265/864) were also up-regulated in *Tdp43*^oKO embryos (Fig. 2d). The overlapped down-regulation 384 genes were involved in ribonucleoprotein complex biogenesis, ncRNA metabolic process, ribosome biogenesis, and rRNA processing, including 20.1% (77/384) of major ZGA genes, such as *Nt5c*, *Itgb5*, *Gja1*, *Mars2*, *Gldc*, *Npl*, *Ttyh2*, *Surf2*, *Lst1*, *Socs3*, *Ptrh1*, *Rusc1*, and *Ctsa* (Fig. 2e, f). The overlapped up-regulation 265 genes were enriched in gland development and regulation of epithelial cell proliferation, including numerous minor ZGA genes, such as *Zscan4a/c/f*, *Zfp352*, *Usp17la/c*, and *Defb13* (Fig. 2e, Supplementary Fig. 4h). The transcriptome profile of *Tdp43*^oKO and KD L2C embryos were more similar to control E2C embryos (Supplementary Fig. 4c, d). Collectively, these results suggest that TDP-43 is required for mouse ZGA during early embryonic development.

## TDP-43 interacts and co-occupies with Pol II as large foci at 2-cell embryos

We observed that TDP-43 translocates from the cytoplasm into the nuclei at the early 2-cell stage, coinciding with the Pol II transition from minor ZGA to major ZGA[7]. Depletion of maternal TDP-43 resulted in the defect of ZGA. These observations raise a possibility that TDP-43 might regulate the ZGA through Pol II in mouse early embryos. We stained the largest Pol II subunit Polr2a in control and *Tdp43*^oKO L2C embryos. Compared with those in the control, the signal of Polr2a staining was decreased in *Tdp43*^oKO L2C embryos, but there was no

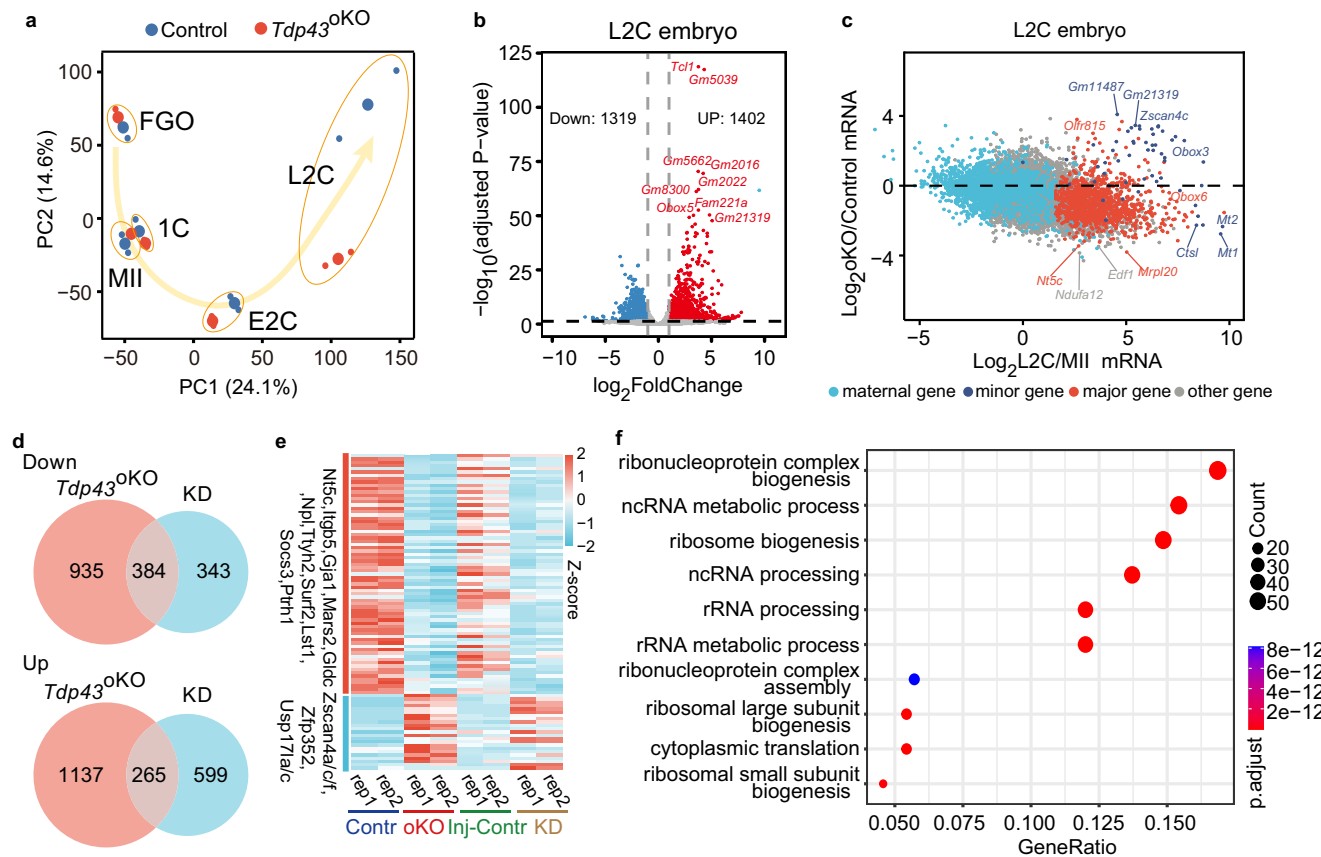

**Fig. 2 | Disruption of maternal *Tdp43* leads to abnormal occurrence of ZGA in mouse two-cell embryos. a** Principal component analysis for RNA-seq data of FGO (full-grown oocytes), MII (MII oocytes), 1 C (late 1-cell embryos), E2C (early 2-cell embryos) and L2C (late 2-cell embryos) from control and *Tdp43*^oKO^ females. **b** Volcano plots showing the differently expressed genes (DEGs) between control and *Tdp43*^oKO^ L2C. 1402 and 1319 representing the number of the genes with significantly up- and down-regulated expression in *Tdp43*^oKO^ L2C compared with the control L2C (adjusted by *P*-value ≤ 0.05; fold change ≥ 2). *P*-values were calculated by Wald significance tests and adjusted with Benjamini and Hochberg method with DESeq2. **c** Scatter plot showing mRNA fold-change (log 2 ratio) between control and *Tdp43*^oKO^ group (oKO) of L2C (*Y*-axis), as well as mRNA fold-change between

MII and L2C (*X*-axis). Light blue, dark blue, red, and gray dots delineate maternal, minor ZGA, major ZGA, and other genes, respectively. **d** Venn diagram showing the number of overlapped genes with the down- and up-regulated expression in both L2C group (*Tdp43*^oKO^, oKO; KD group, knock-down by the Trim-away with TDP-43 antibody and the embryos injected IgG and Trim21 were used as the Injection control, Inj-Contr). **e** Heat map showing the ZGA genes affected in both *Tdp43*^oKO^ and KD L2C embryos. rep, repeat sample. **f** GO terms of the down-regulated genes that were affected in both *Tdp43*^oKO^ (**b**) and KD L2C embryos (Fig. S4e). *P*-values were calculated through one-sided hypergeometric test. The size of dots represents the number of specific genes.

significant change in its mRNA in these embryos (Supplementary Fig. 5a–d). Phosphorylation of the C-terminal domain (CTD) of Polr2a is important for the coordination of transcription events and the phosphorylation at Ser2 (P-Ser2) or Ser5 (P-Ser5) of CTD regulates Pol II undergoing elongation from pausing or involving the initiation of transcription[36,37]. The signal of P-Ser5 was diffused in the nucleus and was comparable between control and *Tdp43*^oKO^ L2C embryos (Supplementary Fig. 5e, f). Interestingly, TDP-43 and P-Ser2 partially co-localized in the nucleus as large foci in the control embryos (Fig. 3a), especially for the embryos treated with PFA and Triton X-100 simultaneously (Fig. 3b). TDP-43 and P-Ser2 were well co-localized as the foci in the embryos with simultaneous treatment of PFA and Triton X-100 (Fig. 3b). These large foci of P-ser2 were, however, lost in *Tdp43*^oKO^ L2C embryos (Fig. 3a). Such defect was only observed in *Tdp43*^oKO^ embryos, but not in *Tdp43*^oKO^ oocytes (Supplementary Fig. 5g, h).

To investigate the direct regulation of TDP-43 in Pol II, we performed Co-immunoprecipitation (Co-IP) experiments in mouse ESCs with the antibodies against TDP-43 and Polr2a. Our results showed that the TDP-43 antibody specifically precipitated Polr2a (Fig. 3c, left panel), even under stringently washing conditions (washed with 500 mM NaCl) (Supplementary Fig. 5i). Conversely, Co-IP experiments

showed that Polr2a could also pull down TDP-43 (Fig. 3c, right panel). The interaction between TDP-43 and Polr2a was confirmed in mouse 2-cell embryos by Proximity Ligation Assay (PLA), a method in situ to detect protein interactions[38] (Fig. 3d). Another negative control was performed by using the antibody against TDP-43 and Tead1 (TEA domain transcription factor 1) to support the specific interaction of PLA in mouse early embryos (Supplementary Fig. 5j). The interaction between TDP-43 and Polr2a by the PLA could be detected in the nuclei of blastomeres as early as at E2C stage in control, but not in *Tdp43*^oKO^ embryos (Fig. 3e). Similarly, TDP-43 was also observed to interact with P-Ser2 of Pol II in the nuclei of blastomeres from the stage of E2C embryo by the PLA (Fig. 3f, Supplementary Fig. 5k). Altogether, these results suggest that TDP-43 is involved in mouse ZGA through interacting with Pol II.

**TDP-43 and RNA Pol II co-occupy at the promoters of ZGA genes**
We then examined whether TDP-43 co-occupies chromatin with Pol II by Stacc-seq technology that we recently established for the samples of scarce cells, with a similar strategy to CUT&Tag[7]. The Stacc-seq data of TDP-43 were highly reproducible between the replicates, and were distinct between FGO and L2C embryos, with only 8.9% (10,313/115,291) of TDP-43 peaks overlapping (Fig. 4a, Supplementary Fig. 6a),

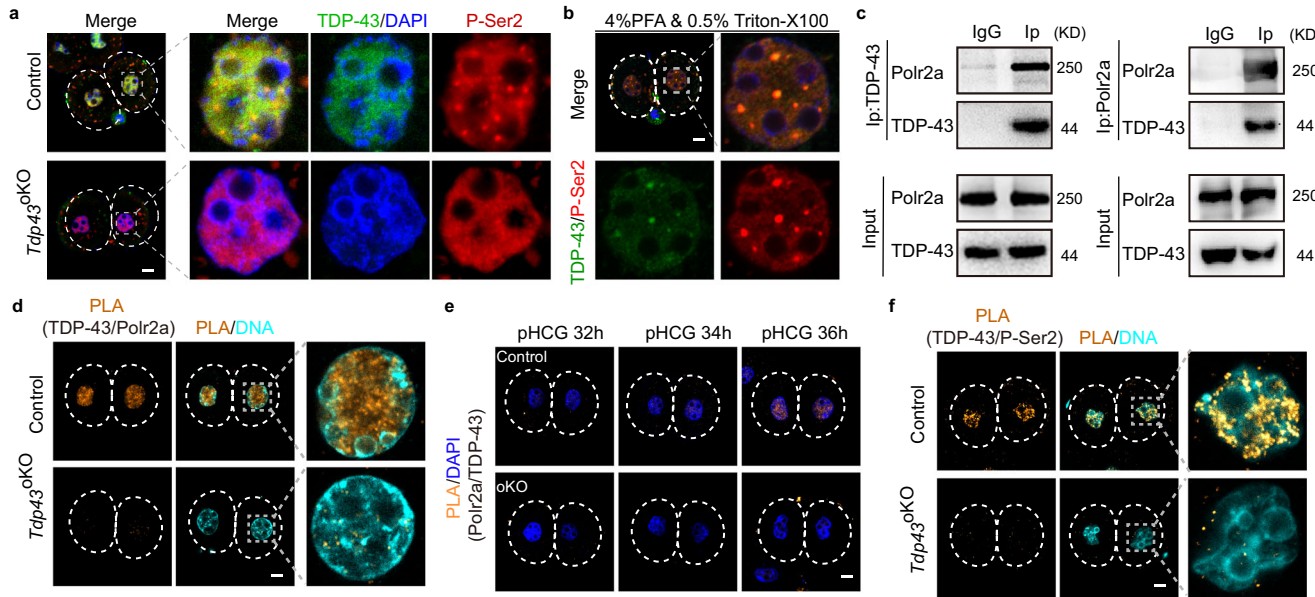

**Fig. 3 | TDP-43 interacts with Pol II in mouse two-cell embryos.**
**a** Immunostaining of the phosphorylated Ser2 of C-terminal domain of Polr2a subunit (P-Ser2) in normal control and *Tdp43*^oKO L2C (late two-cell embryos, *n* ≥ 30). The enlarged pictures of one nucleus of 2-cell embryo were shown on the right. Scale bar, 10 μm. **b** Immunostaining of P-Ser2 and TDP-43 in normal L2C (*n* ≥ 22) after the simultaneous treatment with 4% PFA and 0.5% Triton X-100 for 25 min, Scale bar, 10 μm. **c** Western blotting of TDP-43 and Polr2a for Co-immunoprecipitation (Co-IP) products. Mouse embryonic stem cell lysis was Co-precipitated with TDP-43 or Polr2a antibody. The Co-IP products and input were performed for the Western blotting with specific antibodies. Normal rabbit IgG was used as negative control. **d** Proximity ligation assay (PLA) showing the interaction between TDP-43 and Polr2a in control and *Tdp43*^oKO L2C. L2C (*n* ≥ 24) were isolated from control and *Tdp43*^oKO females and performed for the PLA with TDP-43 and Polr2a antibody (Materials and Methods). Scale bar, 10 μm. **e** PLA results of TDP-43 and Polr2a in control and *Tdp43*^oKO (oKO) early 2-cell embryos at different time points (*n* ≥ 15). pHCG post HCG, h hours. Scale bar, 10 μm. **f** PLA results of TDP-43 and P-Ser2 in control and *Tdp43*^oKO L2C embryos (*n* ≥ 25). Scale bar, 10 μm.

indicating that its binding was stage-specific. To test whether TDP-43 distribution was bona fide in these samples, we analyzed the Stacc-seq data of 1-cell embryos, in which TDP-43 was specifically absent in the nucleus (Fig. 1a). The results showed that there was no significant enrichment of TDP-43 in the genome of 1-cell embryos compared with those in the FGO and L2C embryos (Supplementary Fig. 6b, c). Furthermore, a similar distribution of TDP-43 in FGO with an additional TDP-43 antibody was observed, and the TDP-43 distributions detected by these two antibodies were highly reproducible (Materials and Methods, Supplementary Fig. 6b, d). In addition, we compared TDP-43 distribution with open chromatin that may give rise to no-specific binding for Stacc-seq. The results showed that ~14% or ~28% peaks of TDP-43 overlapped with open chromatin peaks in FGO or L2C, respectively, suggesting that TDP-43 binding was globally distinct from open chromatin (Supplementary Fig. 6e)[39,40]. These data collectively suggest that the TDP-43 distributions of Stacc-seq in the oocytes and embryos are bona fide.

Globally, the stage-specific binding of TDP-43 was well correlated with the dynamics of Pol II at corresponding stages (Fig. 4a, Supplementary Fig. 6f). In addition, TDP-43 significantly enriched at the promoters (35%) in L2C embryos, compared with those in FGO (17%) or in random regions (6%) (Supplementary Fig. 6g). TDP-43 also preferentially occupied some 2-cell specific repeats, such as Alu, Sine B2, and Mervl (Supplementary Fig. 6h), but not LINE1 as reported in the cleavage stage embryos[41].

Next, we focused on the specific binding of TDP-43 and Pol II in L2C embryos. ~78.2% (19,782/25,282) of TDP-43 peaks co-localized with the Pol II peaks in L2C embryos (Fig. 4b, c), while ~21.8% (5,500/25,282) of TDP-43 peaks were specific (Fig. 4b). The overlapped peaks were enriched at the promoters (41%) (Supplementary Fig. 6i), and these promoters were CG-rich (Fig. 4d). TDP-43 and Pol II co-bound the promoters of ~66% (773/1,172) of ZGA genes that tended to function in protein transport, mRNA processing, and RNA splicing (Fig. 4e,

Supplementary Fig. 6j). Both TDP-43 and Pol II were enriched around the TSS of major ZGA genes, but not of minor ZGA genes in L2C embryos (Fig. 4c, f), indicating that TDP-43 is required for major ZGA. The TDP-43-Pol II co-binding peaks were also positively correlated with the regions of open chromatin[40], active histone modifications H3K4me3[42] and H3K27ac[43] (Fig. 4b), indicating that TDP-43 and Pol II co-binding at the promoters of ZGA genes is correlated with active transcription in mouse ZGA.

## TDP-43 regulates the expression of ZGA genes through RNA Pol II

The defective ZGA and the disruption of P-ser2 foci in *Tdp43*^oKO embryos prompted us to investigate whether TDP-43 depletion could affect the transition of Pol II occupancy during ZGA. By applying Stacc-seq[7], we found that the change of Pol II occupancy at the promoters and gene bodies was correlated with the expression levels of DEGs in *Tdp43*^oKO L2C embryos (Supplementary Fig. 7a, b). We separated promoter and distal Pol II peaks in the control and *Tdp43*^oKO embryos into three clusters according to our previous report[7], including the shared cluster between 1-cell (1 C, PN5) and late 2-cell (L2C), as well as the 1 C and L2C specific cluster. The PoI II enrichment of the 1 C specific cluster became gradually weaker, while the enrichment of the L2C specific cluster became gradually stronger in WT embryos (Fig. 5a).

Compared with those in control L2C embryos, although the shared and L2C specific PoI II enrichment was not obviously varied in *Tdp43*^oKO L2C embryos, the 1 C specific enrichment was slightly stronger at both promoters and distal regions (Fig. 5a), suggesting the involvement of TDP-43 in Pol II transition from minor to major ZGA, in coordination with the time of TDP-43 accumulating to the nucleus. Consistently, compared with control, Pol II was more enriched at minor ZGA genes (Fig. 5b, c, Supplementary Fig. 7c, d), but less enriched at major ZGA genes in *Tdp43*^oKO L2C embryos, especially at the gene bodies (Fig. 5d–f, Supplementary Fig. 7e). In addition, compared

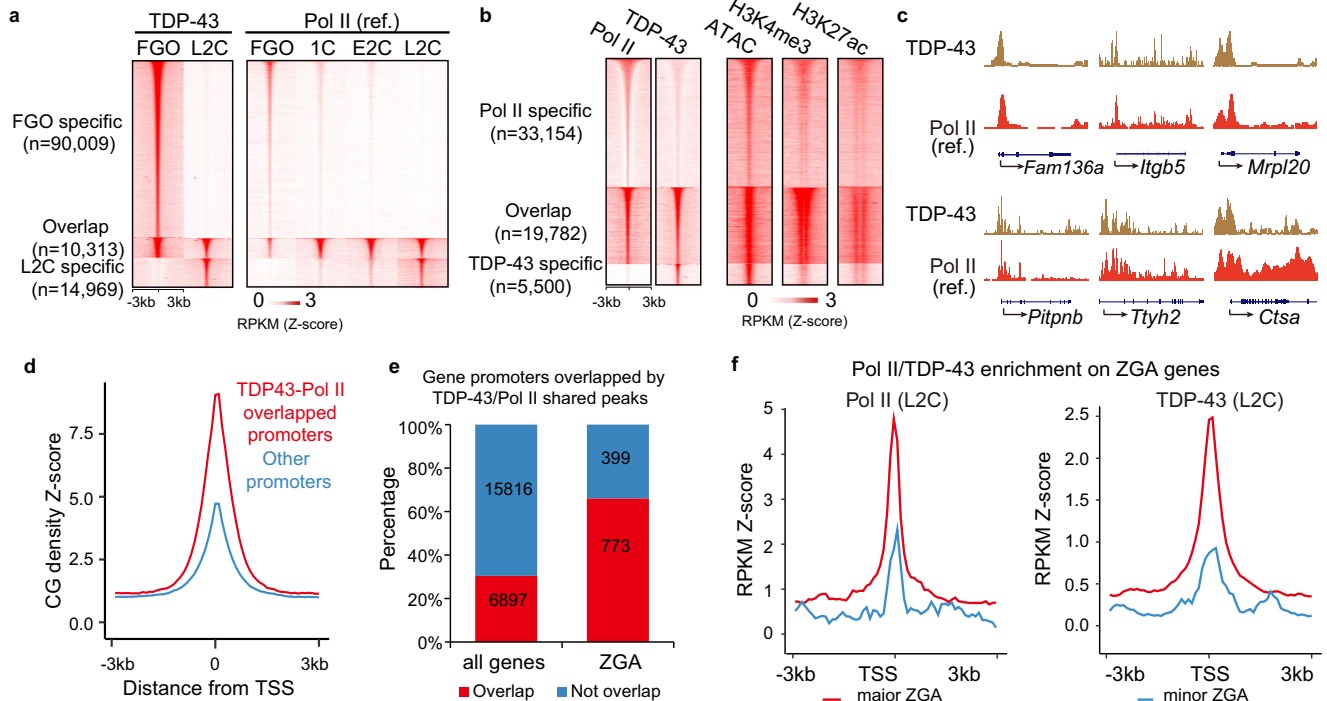

**Fig. 4 | TDP-43 and Pol II co-occupy at mouse ZGA genes. a** Heat maps showing the enrichment of TDP-43 at normal FGO and L2C specific peaks (left panel), as well as the overlapped PoI II enrichment peaks at wild type mouse oocytes and early embryos (right panel). Pol II enrichment (ref.) was referred from Liu, et al. 2020. FGO, full-grown oocyte; 1 C, 1-cell embryos; E2C, early 2-cell embryos; L2C, late 2-cell embryo. Z-score was normalized. **b** Heat maps showing the specific and overlapped enrichment of Pol II and TDP-43 (left panel), as well as their ATAC (chromatin accessibility) (Ref., Wu, J. et al. 2016), H3K4me3 (Ref., Zhang, B. et al. 2016) and H3K27ac enrichment (Ref., Dahl, J. A. et al. 2016) (right panel) in normal L2C. **c** Snapshots showing TDP-43 and Pol II (Ref., Liu, et al. 2020) enrichment in normal L2C at representative major ZGA genes. **d** The CG density of promoters covered by both of TDP-43 and Pol II. Z-score was normalized. **e** Histograms showing the percentage of the promoters occupied by both TDP-43 and Pol II. The overlapping and not overlapping promoters accounted for the proportion of all genes and ZGA genes. **f** The enrichment of Pol II (left) and TDP-43 (right) at the promoters of major and minor ZGA genes in L2C. RPKM was normalized with Z-score. TSS, transcription start site.

with control, Pol II maintained a high occupation at Mervl in *Tdp43*[oKO] L2C embryos (Fig. 5c), which is usually activated at mouse early 2-cell stage and shut down at late 2-cell stage[7]. These observations were consistent with the increased and decreased expression of minor and major ZGA genes in *Tdp43*[oKO] L2C embryos, respectively (Fig. 2c, e). Collectively, these results suggest that Pol II transition from minor ZGA to major ZGA is aberrant in *Tdp43*[oKO] L2C embryos.

Next, we sought to test whether gene expression changes in *Tdp43*[oKO] L2C embryos were related to TDP-43 binding. Globally, the enrichment of TDP-43 at the promoters of down-regulated genes upon TDP-43 knockout was higher than those at other promoters (Fig. 6a), suggesting the down-regulated genes were more likely to be regulated by TDP-43 directly. We also separated all genes into the unbound and other four categories according to the binding enrichment of TDP-43 at their promoters (Supplementary Fig. 7f). Consistently, the genes bound more by TDP-43 at their promoters were more likely to be down-regulated upon the loss of TDP-43 (Fig. 6b, c), and these genes might maintain a relatively high expression level from 2-cell until 8-cell embryos[42] (Supplementary Fig. 7g). However, the DEG expression and Pol II enrichment were not related to the binding of TDP-43 in the oocytes with TDP-43 deletion (Supplementary Fig. 8a–d). Hence, these data suggest that TDP-43 co-occupies the ZGA genes with Pol II and is directly involved in the expression of ZGA genes in mouse early embryos.

**TDP-43 interacts with Cyclin T1**

Our results suggest that TDP-43 was directly involved in the transcription of ZGA genes through Pol II configuration. As mentioned previously, Pol II enters the elongation for transcript production when it is phosphorylated at its CTD Ser2 (P-Ser2) by P-TEFb (the positive transcription elongation factor complex, consisting of Cyclin T1/2 and CDK9, encoded by *Ccnt1/2* and *Cdk9*)[36,37]. Then, we sought to investigate whether TDP-43 regulates Pol II through the components of the P-TEFb complex in mouse ZGA.

CDK9 localized in nuclei from the 1-cell stage (Supplementary Fig. 8e), probably requiring for Cyclin T2 function in mouse pronucleus to initiate minor ZGA[44]. Similar to TDP-43 (Fig. 1a, b), Cyclin T1 was reported to appear in the nucleus at the two-cell stage[45]. Therefore, we performed Co-IP experiments with *Myc-Ccnt1* and *Flag-Tdp43* vectors in 293 T cells. The results showed that TDP-43 interacted with Cyclin T1 (Fig. 6d, e). PLA results suggested that the endogenous interaction between TDP-43 and Cyclin T1 was detected in mouse L2C embryos (Fig. 6f). The interplay between TDP-43 and Cyclin T1 could be detected through PLA since the early 2-embryo stage when TDP-43 was accumulated in the nucleus (Fig. 6g), further supporting their endogenous interaction. In addition, exogenous EGFP-Cyclin T1 was colocalized with the foci of P-Ser2 (Supplementary Fig. 8f). Taken together, these data suggest that TDP-43 is involved in Pol II regulation through interacting with Ccnt1 of P-TEFb during mouse zygotic genome activation.

## Discussion

Zygotic genome activation generates transcripts to replace the maternal supply and to initiate the development of embryos[4]. In the present study, we have found that TDP-43 translocates from the cytoplasm to the nucleus, co-occurring with the beginning of major ZGA at the mouse early 2-cell stage[31]. TDP-43 interacts and co-occupies with Pol II at 66% of promoters of mouse ZGA genes. Furthermore, we

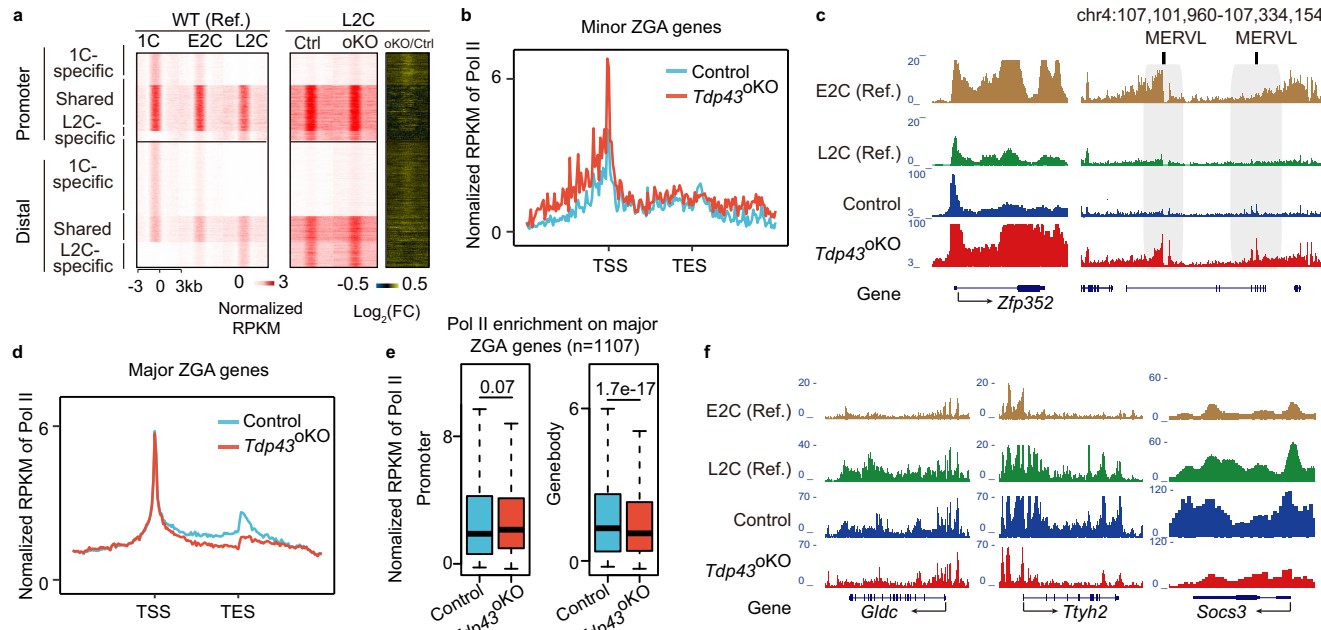

**Fig. 5 | Deletion of maternal *Tdp43* leads to incorrect configuration of Pol II.**
**a** Heat maps showing the shared and stage-specific enrichment of Pol II at the promoter and distal regions in WT (Ref.) and L2C embryos. Left panel was WT (Ref.) from Liu, et al. 2020. WT wild type, 1 C 1-cell embryo, E2C and L2C, early and late 2-cell embryo; *Ctrl* control; *oKO* oocyte-specific knockout. **b** Line chart showing the enrichment of Pol II at minor ZGA genes in the control and *Tdp43*^oKO L2C. Z-score was normalized. TSS, transcription start site; TES, transcription end site. **c** Snapshot showing Pol II enrichment at minor ZGA gene *Zfp352* and Mervl in E2C (Ref.) and L2C (Ref.), Control and *Tdp43*^oKO L2C. **d** The enrichment of Pol II at major ZGA genes in control and *Tdp43*^oKO L2C. **e** The box plot showed the enrichment of Pol II at promoters and gene bodies of major ZGA genes in Control and *Tdp43*^oKO L2C. Tests were run with two-sided Wilcoxon rank-sum. Upper and lower line of the box representing the 25th and 75th percentiles. Whiskers showing 1.5X inter-quartile range. Central line, the median. **f** Snapshot showing Pol II enrichment at major ZGA gene *Gldc*, *Ttyh2*, *Socs3* in E2C (Ref.) and L2C (Ref.), as well as Control and *Tdp43*^oKO L2C.

find that TDP-43 interacts with Cyclin T1 of the positive transcription elongation factor complex P-TEFb during mouse ZGA. Through antagonizing the negative elongation factors, Cyclin T1 phosphorylates Ser2 of Pol II CTD to promote Pol II escaping from the promoter pausing to enter the elongation[37,46,47]. Thus, the translocation of TDP-43 from the cytoplasm to the nucleus may assist the release of Pol II from the promoter pausing to activate the transcription of the zygotic genome by recruiting Cyclin T1. Consistent with this hypothesis, disruption of maternal TDP-43 results in the decreased expression of ZGA genes and reduced enrichment of Pol II at the bodies of major ZGA genes and loss of P-ser2 foci in mouse two-cell embryos with the depletion of TDP-43. The co-localization of TDP-43, Pol II P-ser2 and Cyclin T1 in the large foci in mouse two-cell embryos further supports the hypothesis. Taken together, these lines of evidence strongly support that TDP-43 promotes the expression of ZGA genes by activating transcription of RNA Pol II elongation from its pausing through Cyclin T1 during mouse maternal-to-zygotic transition.

During oocyte to embryo transition, many nuclear factors should be released from the nucleus of mouse oocytes after the germinal vesicle breaking down and be relocated into the nucleus after fertilization for their functions. Previously, Tif1α was reported to enter the pronucleus at the mid and late 1-cell embryo stages when minor ZGA occurs and to be required for mouse minor ZGA[48]. Cyclin T1 was reported to translocate from the cytoplasm into the nucleus at the 2-cell stage and to be related to mouse ZGA[7,45]. H2A.Z was reported to be gradually deposited in the nucleus during ZGA in *Drosophila*[49]. Recently, the maturation of nuclear pore complex in zebrafish early embryos was reported to control nuclear factor transportation and the correct ZGA timing[50]. Although these nuclear factors have been shown to be involved in the ZGA, how these factors are transported into the nucleus remains unclear. Here, we have found that TDP-43 translocates from the cytoplasm to the nucleus during the transition of mouse ZGA.

The exogenous TDP-43 with nuclear localization sequence could recruit the endogenous Cyclin T1 into the nucleus of mouse 1-cell embryos (Supplementary Fig. 8g, h). Thus, TDP-43 translocation may regulate the function of Cyclin T1 in Pol II elongation during mouse ZGA. Therefore, the precise importation of key nuclear proteins is probably a key mechanism to regulate the zygotic genome activation during the beginning of animal life.

Since the discovery of TDP-43 aggregations related to neurodegenerative diseases, the translocation of TDP-43 from the cytoplasm into the nucleus has been broadly investigated in the neuronal cells with cytoplasmic TDP-43 aggregations for the treatment of these diseases[51]. However, the regulations of TDP-43 translocation remain intangible. TDP-43 contains a classic NLS that has been shown to contact importin α1/β and promotes TDP-43 translocation from the cytoplasm TDP-43 aggregation into the nucleus[52]. The importin α (also known as karyopherin α) family proteins include six members that highly express during mouse oocyte to embryo transition[53]. Maternal importin α2 (Kpna7) was reported to be required for mouse early embryonic development and fecundity[54]. Similar to those lacking maternal TDP-43, depletion of maternal importin α7 (Kpna6) resulted in the arrested embryo development at the two-cell stage and the dramatically decreased expression of ZGA genes in these embryos[53]. Thus, these importin proteins may interact with TDP-43 and regulate its translocation from the cytoplasm into the nucleus during mouse maternal to zygotic transition. Taking together, the regulations of TDP-43 translocation deserve to be further investigated during mouse maternal to zygotic transition. The investigations of TDP-43 translocation may contribute to the treatment of human neurodegenerative diseases, such as amyotrophic lateral sclerosis and Alzheimer's disease.

Although TDP-43 co-localizes with Pol II in mouse full-grown oocytes, the absence of TDP-43 barely affects P-ser2 foci formation and

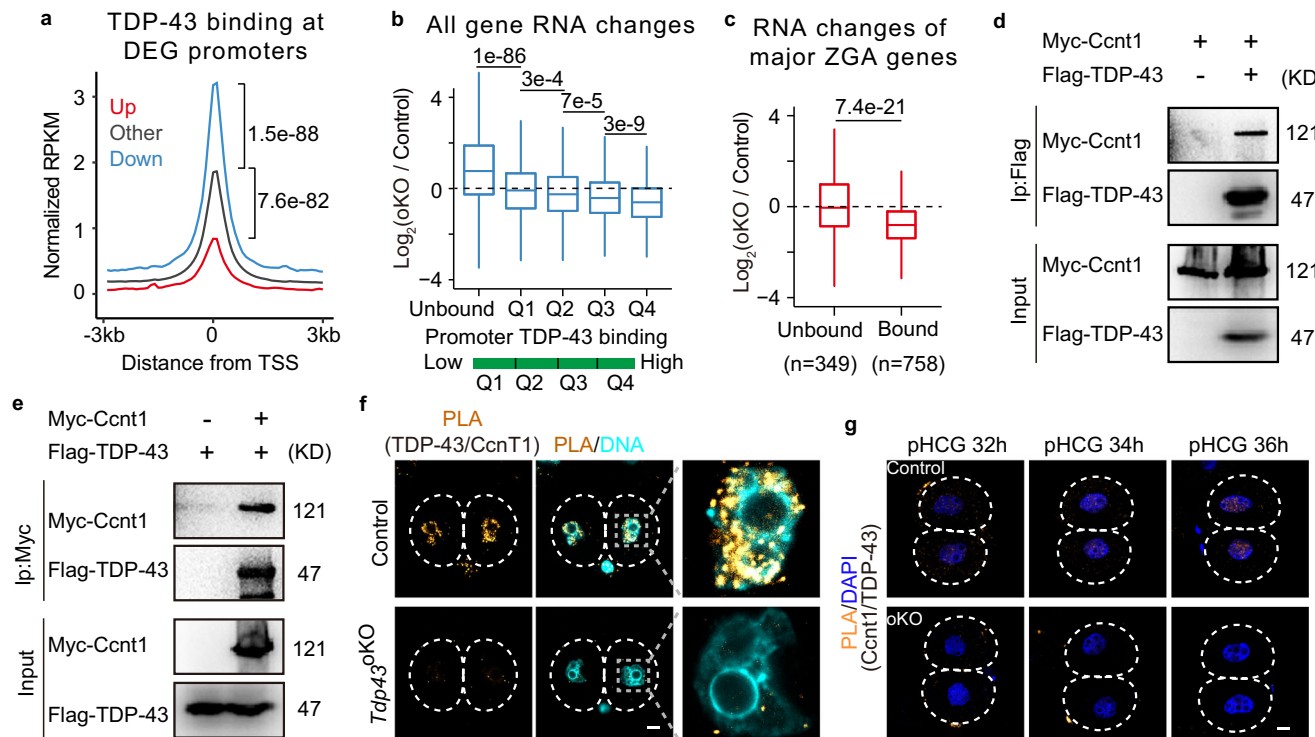

**Fig. 6 | The effect of TDP-43 binding on gene expression in L2C and the interaction between TDP-43 and Cyclin T1. a** The enrichment of TDP-43 at the promoter of DEGs identified in control and *Tdp43*oKO late two-cell embryos. All tests were run with two-sided Wilcoxon rank-sum. *P*-value were shown. DEGs, different expression genes; RPKM, reads per kilobase million; Up, Down, and Other, up-, down-regulated and other genes; TSS, transcription start site. **b** The box plots showing the mRNA fold-changes (log 2 ratio) for all genes at the promoters in control and *Tdp43*oKO (oKO) L2C. The promoters were characterized as Supplementary Fig. 7f (Unbound, 15325 genes; Q1, 1843 genes; Q2, 1847 genes, Q3, 1850 genes; Q4, 1848 genes). Upper and lower line of the box representing the 25th and 75th percentiles. Whiskers showing 1.5X interquartile range. Central line, the median. The tests were run with the two-sided Wilcoxon rank-sum. **c** The box plots

showing the mRNA fold-changes (log 2 ratio) for major ZGA genes at the promoters in control and *Tdp43*oKO (oKO) L2C. The promoters of major ZGA genes were characterized by unbound or bound TDP-43. Upper and lower line of the box represent the 25th and 75th percentiles. Whiskers show 1.5X interquartile range. Central lines represent the median. The test was run with the two-sided Wilcoxon rank-sum. **d**, **e** Western blotting of the Co-IP products with Flag (**d**) or Myc (**e**) antibody. 293t cells were transfected with different vectors for Co-IP experiments. Flag-TDP-43, *TDP-43* labeled with *Flag*; Myc-Ccnt1, *Ccnt1* labeled with *Myc* and *EGFP*. **f** The PLA for the interaction between TDP-43 and Ccnt1 detected with the specific antibodies in control and *Tdp43*oKO L2C (*n* ≥ 22). Scale bar, 10 μm. **g** The PLA for the interaction between TDP-43 and Ccnt1 in control and oKO (*Tdp43*oKO) E2C embryos (*n* ≥ 15) at different time points. pHCG post HCG, h hours. Scale bar, 10 μm.

Pol II configuration in these cells. Furthermore, the absence of TDP-43 mildly affects the gene expression in mouse full-grown oocytes. These observations suggest that the regulation of TDP-43 on the active transcription through Pol II is a stage- or context-specific function of ZGA during mouse oocyte to embryo transition. A previous report shows that TDP-43 regulates the expression of a testis-specific *Acrv1* (acrosomal vesicle protein 1) gene during mouse spermatogenesis, in which TDP-43 and Pol II binds the promoter of *Acrv1* in mouse spermatocytes and round spermatids, but the expression of this gene is exclusive in the post-meiotic round spermatids[22], further supporting this hypothesis. Thus, how the cells control TDP-43 for the active transcription in the specific context deserves to be further investigated. In summary, our study provides strong evidence to support that maternal TDP-43 interacts with Pol II and is required for proper Pol II configuration, ZGA, and early embryogenesis.

## Methods
### Mice
Mice and all animal experiments were manipulated under the guidelines of the Animal Care and Use Committee of the Institute of Zoology, Chinese Academy of Sciences (IOZ-IACUC-2021-052). Mice were maintained under the controlled environment conditions, including 12-h alternating light/dark cycles, a temperature controlled at 23-25 °C, and a humidity range of 40–65%. They had ad libitum access to food and water. *Tdp43*floxp/floxp mice (No. 017591) that possess loxP sites

flanking exon 3 of *Tdp43* were obtained from Jackson Laboratory and were mated with *Zp3-Cre* transgenic mice that express *Zp3* promoter driving Cre recombinase[33]. After obtaining *Tdp43*+/floxp;*Zp3-Cre* mice, they were intercrossed to produce *Tdp43*floxp/floxp;*Zp3-Cre* and other genotype mice. *Tdp43*flox/flox females were mated with *Tdp43*flox/flox;Zp3-Cre males to routinely produce *Tdp43*floxp/floxp (female as control) and *Tdp-43*floxp/floxp;*Zp3-Cre* mice. *Tdp43* floxp/floxp;*Zp3-Cre* females had the oocytes with TDP-43 deletion, thus designed as *Tdp43*oKO (*Tdp43* oocyte-specific knockout) mice. These mice were maintained in a hybrid background of C57BL/6 J and ICR, and were genotyped using tail DNA or toe DNA as PCR templates with the primers of a previous report (PubMed: 20660762)[33].

### Collection of mouse oocytes and embryos
Oocytes and embryos were isolated in M2 medium from 7–8-week-old female mice that were injected with HCG after the injection of PMSG for 46–48 h or mating with males at the following time points: MII oocytes: 13 h after HCG; PN2 1-cell embryos: 24 h after HCG; PN5 1-cell embryos (Late 1-cell): 30 h after HCG; early two-cell embryos: 36 h after HCG; late two-cell embryos: 48 h after HCG. To obtain embryos, females injected with HCG were mated with normal fertile ICR males. Full-grown oocytes were isolated from the ovaries of 7–8-week-old females after only the injection of PMSG for ~48 h and cultured in M2 medium (Sigma 7167). Embryos were cultured in KSOM medium (Millipore MR-106-D).

## Quantitative real-time RT-PCR

Oocytes and embryos were transferred to the lysis buffer of Invitrogen Dynabeads™ mRNA DIRECT™ Micro Kit (Millipore 61021). mRNA was extracted as the instruction of the manufacturer. Reverse transcription (RT) was performed with Takara PrimeScript RT Reagent Kit according to the instruction of the manufacturer. Quantitative RT-PCR was conducted with Abm EvaGreen 2X qPCR MasterMix-No dye kit (Abm MasterMix-S) with specific primers of *TDP-43* (*GCTGGCTGGGGCAATCTG, CAGGGGAGACCCAACACTATG*) and *Gapdh* (*ACCACAGTCCATGCCATCAC, TCCACCACCCTGTTGCTGTA*). The results were repeated with at least three independent samples. *Gapdh* was used to normalize the gene expression.

## Immunofluorescence staining

Oocytes and embryos were fixed with 4% PFA in PBS for 30 min at 37 °C. These samples were washed with PBSA (0.1% BSA, about 5 min each time), permeabilized with 0.5% (V/V) Triton X-100 for 30 min at room temperature. After being washed with PBSA, samples were blocked with 1% BSA for 1 h at room temperature. These samples were incubated with primary antibody overnight at 4 ˚C (TDP-43 antibody-1, Proteintech 12892-1-AP, lot 00043028 & lot1, 1:300; TDP-43 antibody-2, Santa Cruz Sc-376532, lot #Il014, 1:100; P-ser2, Abcam ab5095, lot GR3195689-1 & lot GR3353130-2, 1:250; P-ser5, Abcam ab193467, lot GR3298885-1, 1:200; Polr2a, Affinity DF6831, lot 83s1054, 1:200; Ccnt1, Abcam 184703, lot GR250586-7, 1:100; Tead1: Cell signaling Technology 12292 S, lot 3, 1:200; CDK9, Abcam ab76320, lot GR3439647-3, 1:200). After washed with PBST (0.1% triton X-100 in PBS) three times, oocytes and embryos were incubated with specific fluorescence conjugated secondary antibody for 1 h at room temperature (all the secondary antibodies were from Jackson Laboratory). After washed at least three times, the samples were imaged with Zeiss LSM780 or LSM880 or Airyscan880. For TDP-43 staining (Santa Cruz Sc-376532), the zona pellucida of embryos was removed with Tyrode's solution (Sigma T1788) before fixation.

## RNA-seq library generation and sequencing

Oocytes and embryos were treated gently with Tyrode's solution for removing the zona pellucida. Twenty oocytes or embryos were collected for each group of RNA extraction. These samples were performed for reverse transcription and PCR pre-amplification as the smart-seq2 steps as previously reported[55]. The RNA library was constructed with a kit of Vazyme (TD502) and performed as the protocol of manufacture.

## RNA-seq data processing and identification of ZGA genes

All RNA-seq data were mapped to the mouse mm9 genome by HISAT2 (v2.2.1) with default parameters after removing the low-quality sequencing reads with Cutadapt (v0.3.8)[56]. The gene expression was calculated with gene annotation from Gencode RefSeq database using HTSeq (v2.2.1), StringTie (v2.1.2) and FeatureCounts (v2.0.1)[57,58]. DEGs were identified by DESeq2 (v1.24.0)[59], and the threshold of adjusted *P*-value was 0.05 with fold change ≥2. Heatmaps of the genes with expression changes were clustered with k-means clustering analysis and were visualized with the heatmap package in R. 2D. PCA analysis was performed and visualized by using all expressed genes by factoextra (v.1.0.7). Dot analysis was used to compare the up- or down-regulated expressed genes in the samples and the Spearman's rank correlation coefficient was used to compare the relevance between different samples. GO term analysis for the enriched biological processes was conducted using clusterProfiler (v.4.6.0) and DAVID (v6.8) with the default parameters[60].

Maternal and ZGA genes were defined with the reference RNA-seq data of mouse oocytes and early embryos[42]. Maternal genes were identified as highly expressed in MII oocytes (FPKM > 5), but down-regulated in late 2-cell embryos (at least 3-fold). Genes with FPKM (MII) < 5 and up-regulation (FPKM > 5, at least 3-fold) in either 1-cell embryos or early 2-cell embryos were identified as minor ZGA genes, while those genes up-regulated in late 2-cell embryos were identified as major ZGA genes. Other genes with FPKM (MII) > 5 and highly up-regulated at the late 2-cell stage (over 5-fold) were also included in major ZGA genes.

## In vitro transcription and microinjection of 1-cell embryos

*Trim21, NLS-TDP43* and *Ccnt1* were cloned in the vector Pcs2 that contained a fused-expressing mCherry or EGFP. The cloned genes were PCR with the sequence of SP6 promoter and T3 promoter. The cloned genes were transcribed according to the AM1340 manufacturer's protocol and the products were purified as the mRNA template for the microinjection of mouse 1-cell embryos before PN2 stage. After injected with *NLS-TDP43-mCherry* and *Ccnt1-EGFP*, mouse 1-cell embryos were cultured for these gene expression and fixed for the staining of TDP43 and Ccnt1 with specific antibodies. For Trim-away technology, *mCherry-Trim21* mRNA and rabbit anti-TDP-43 antibody (Proteintech,12892-1-AP) or normal rabbit IgG were simultaneously injected into mouse 1-cell embryos before the PN2 stage, then these embryos were cultured in KOSM to develop into late two-cell stage (pHCG 48 h) for the following analysis.

## Cell culture and vector transfection

ESCs were maintained in classical 2i/lif culture. The induced-knock-out(iKO) ESCs were derived from the blastocysts that were isolated from the mating of *Tdp43 floxp/floxp* and ER-cre mice, and were treated with Tamoxifen for 72 h for the immunoprecipitation experiments (washing with NP-40 containing 500 nM NaCl). 293t cells were maintained in classical DMEM basic supplement with 10% FBS.

Vector and transfection reagent PEI (three times of vector molar volume) were dissolved in isopyknic opti-MEM for 5 min and were mixed. Twenty-five minutes later, the mixture of vectors and PEI was added into 293t cells to incubate for 12 h. Then, the culture medium was replaced with the fresh medium. After 36 h of transfection, these cells were collected with the lysis buffer of Beyotime plus protease inhibitor for the immunoprecipitation (IP) experiments.

## Co-immunoprecipitation and western blotting

Mouse ESCs or 293t cells were collected in IP lysis buffer of Beyotime with protease inhibitor, incubated on ice for 15 min. After centrifuging (11833 g for 15 min at 4 °C), the supernatant was collected for Co-IP experiments according to the manufacturer's protocol of Protein A/G Magnetic Beads (Bimake B23202 or MCE HY-K0202). The antibodies included TDP-43 (Proteintech 18280-1-AP, lot 00025213, 1:400), Polr2a (Abclonal A2107, lot 3561768005, 1:300), Flag (Sigma F1804, Clone/M2, 1:400), and Myc (Abmart M20002M, lot 324572, 1:400). We also washed the IP mixture of mouse iKO ESCs with NP-40 containing 500 mM NaCl for the Co-IP experiment of TDP-43. The precipitated products were heated for 5 min at 95 °C for Western blotting.

Oocytes and embryos were collected in 10 μl 1X protein loading buffer and heated for 5 min at 95 °C. The heated samples were fractionated at 100 V for 2 h using 10–12% NuPAGE Bis-Tris gels and MOPS SDS running buffer, and were transferred to PVDF membranes for 2 h at 200 mA (3.5 h when Polr2a antibody was blotted). Then, the membranes were blocked with 5% skimmed milk in TBST (M/V) for 1 h at room temperature. After washed with TBST, the membranes were incubated with primary antibodies at 4 °C overnight (primary antibodies including: TDP-43, Proteintech 18280-1-AP, lot 00025213, 1:1000; Polr2a, Abclonal A2107, lot 356 1768005, 1:1000; Flag, Sigma F1804, Clone/M2, 1:1000; Myc, Abmart, lot 324572, 1:1000; Ccnt1, Abcam,184703, lot GR250586-7, 1:800; GAPDH, Proteintech, 60004-1-lg, lot 10025237, 1:5000). After washed with TBST, the membranes were incubated with HRP-conjugated secondary antibodies at room temperature for 1 h. After washed with TBST, the membranes were

detected for the targeting proteins according to the instruction of SuperSignal West Dura Extended Duration Substrate and were imaged with a LAS-3000 apparatus.

## Proximity ligation assay

Proximity Ligation Assay was performed with Sigma-Aldrich reagents according to the protocol of Sigma Aldrich Duolink® PLA Fluorescence (rabbit Plus probe DUO92002, mouse Minus probe DUO92004, washing buffer DUO82049, Detection reagents DUO92008). The primary antibodies of different host species (mouse anti-TDP-43 antibody: Santa Cruz Sc-376532; rabbit anti-Polr2a: Affinity DF6831; rabbit anti-P-Ser2: Abcam ab5095; and rabbit anti-Ccnt1 antibody: Abcam 184703, rabbit anti-Tead1: Cell signaling Technology 12292 S) were used and diluted as immunofluorescence staining. The embryos were collected as above, and their zona pellucida was removed using Tyrode's solution. These samples were fixed with 4% PFA for 30 min at 37 °C, and permeabilized with 0.5% Triton X-100 in PBS for 30 min at room temperature. The embryos were blocked with Sigma-Aldrich Duolink® Blocking Solution in a humidity chamber for 60 min at 37 °C. The samples were transferred into the mixed solution with the primary antibodies in Duolink® Antibody Diluent, and were incubated in a humidity chamber at 37 °C for at least 2 h or at 4 °C overnight. The samples were washed with 1X Duolink® washing buffer A, and were transported to the probe mixture prepared as the manufacturer's protocol, then were incubated in a humidity chamber at 37 °C for 1 h. After washed with Duolink® washing buffer A, the samples were incubated with the ligation mixture that were prepared as the manufacturer's protocol in a humidity chamber at 37 °C for 30 min. After washed with 1X Duolink® washing buffer A, the samples were incubated with the amplification mixture containing amplification buffer (1:5) diluted with ultrapure water and amplification polymerase (1:80) in a humidity chamber at 37 °C for 100 min. After washed with 1X Duolink® washing buffer B and 0.01X Duolink® washing buffer B, the samples were incubated with 1X DAPI at room temperature for 30 min or at 4 °C for a long time in a dark condition. After washed with PBSA (0.1% BSA in PBS), the samples were imaged by using a LSM880 or Airyscan880 apparatus.

## Stacc-seq library generation and sequencing

Stacc-seq was performed according to our previous report[7]. Briefly, 1 µl fully dissolved 5% digitonin (Sigma D141) was added to Buffer1 (10 mM Tris-HCl pH = 7.4, 150 mM NaCl, 0.5 mM spermidine, 1X EDTA-free Roche complete protease inhibitor), termed DB1. Then, 0.5 µl PA/G-Tn5 (Vazyme Biotech TD902) and 0.5 µg TDP-43 antibody (Proteintech 12892-1-AP, lot1, Diagenode C15410266, lot 43579) or Pol II antibody (Active Motif 102660) were also added to DB1 and the mixture was incubated at 4 °C for 30 min. Meanwhile, mouse oocytes and embryos were treated gently with Tyrode's solution to remove the zona pellucida, then were picked into a 200 µl low-binding tube containing 6 µl of the DB1. After incubated at 4 °C for 10 min, these samples were added with 35 µl of the mixture of DB1, PA/G-Tn5, the antibody and 12.5 µl of TTBL (Vazyme TD502) for cleaving the chromatin around the binding sites and transposing with adapters. Finally, the DNA was purified and PCR was performed as the Stacc-seq steps. We used at least 150 embryos or oocytes for the high quality data of TDP-43 (200) or Pol II (150) in these experiments.

## Stacc-seq data processing

All paired-end Stacc-seq reads were mapped to mm9 using Bowtie2 (v.2.2.5) after trimming adapters and low quality sequencing reads[61]. For downstream analysis, read counts were normalized by computing the numbers of reads per kilobase of bin per million of reads sequenced (RPKM) after removing non-uniquely mapped reads, unmapped reads and PCR duplicates. The RPKM was further normalized with Z-score transformation to minimize the batch effect. UCSC genome browser was used to visualize the signals of Stacc-seq[62]. Repetitive element enrichment was calculated by comparing annotated repeats overlapped by Stacc-seq peaks and randomly shuffled peaks. All the Stacc-seq peaks were called by MACS (v.1.4.2) with the parameters–nolambda–nomodel[63], and were compared by using BEDTools (v2.29.2)[64]. ChIPseeker (v1.26.2) was used to identify and visualize the genomic features of peaks[65]. Promoters were defined as ±2,5 kb around the transcription starting sites (TSS). Stacc-seq peaks at least 2.5 kb away from TSS were defined as distal peaks by BED-Tools (v2.29.2).

## Statistical and reproducibility

Quantitative analyses were expressed as the mean ± SD, based on at least three independent biological samples. Statistical analyses were performed using a two-tailed Student's $t$-test or two-sided Wilcoxon rank-sum to compare the difference between two groups. $P$-values < 0.05 were considered as a significance. The immunostaining experiments and Western blots were performed at least three independent repetitions with similar results.

## Reporting summary

Further information on research design is available in the Nature Portfolio Reporting Summary linked to this article.

## Data availability

All data supporting the findings of this study are available within the paper and its Supplementary information files. All the datasets used in this study are publicly available. The raw and processed data generated in this study have been deposited in GEO with accession number GSE221985. The mm9 mouse reference genome is available at https://hgdownload.soe.ucsc.edu/goldenPath/mm9/bigZips/mm9.fa.gz. Source data are provided with this paper.

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

## Acknowledgements

We thank Drs. Falong Lu, Hu Nie and Rongrong Sun for their help of the sequencing of mouse FGO and MII oocytes. This work was found by the National Key R&D Program of China (2022YFC2702200 and 2021YFC2700300 to L.L., 2021YFA1100102 and 2019YFA0508901 to W.X.), the National Natural Science Foundation of China (31930033 and 32170812 to L.L., 31988101 and 31725018 to W.X.) and the New Cornerstone Foundation (W.X.).

## Author contributions

X.N. and L.L. conceived and designed the project. X.N. performed the majority of the experiments, analyzed the data, and organized the figures. Q.X. and W.X. took charge of the Stacc-seq, advised for some experiments and organized figures, C.X. analyzed the RNA-seq and performed part of Co-immunoprecipitation. F.C. analyzed most of the Stacc-seq data. Q.W. analyzed part of the Pol II Stacc-seq data. D.Q., R.W. and X.Y. helped to collect embryos and oocytes, D.Q. and R.W., involved the injection of the projects. X.L. performed the RNA-seq of the oocytes and embryos from *Tdp43*^okO and control mice. Z.G. optimized PLA and performed some experiments. Y.W. and C.G. prepared some reporters. X.N. wrote the original manuscript. L.L. supervised the project and finalized the manuscript with the comments from all authors.

## Competing interests

The authors declare no competing interests.
