## [Peer Review File · Nature Communications]

Maternal TDP-43 interacts with RNA Pol II and regulates zygotic genome activationEditorial Note: Parts of this Peer Review File have been redacted as indicated to maintain the confidentiality of unpublished data.

REVIEWER COMMENTS

Reviewer #1 (Remarks to the Author):

In this manuscript, Nie et al. explore the role of TDP-43, a DNA binding protein, in early phases of mouse development. They present a maternal TDP-43 knock out mouse with obvious effects on embryo development, and by employing a wide range of genomic and cellular approaches demonstrate the importance of TDP-43 during zygotic genome activation (ZGA). Also, they combine a series of techniques suggesting the interaction of TDP-43 and Pol II, which might be the underlying regulation mechanism. This is a nicely executed work that builds upon previous papers from some of the authors (Liu et al. 2020), and increments our understanding of transcription at the ZGA.

Nevertheless, certain aspects of the manuscript could be improved to transmit a clearer message to the readers, and some new data is required to more firmly support the claims made by the authors. Below follow some general issues, that should be at least discussed and made clear, and more specific issues regarding the experimental data.

GENERAL ISSUES

1. The distribution of TDP-43 in the first stages of development, and the phenotypic consequence of the loss of maternal product, strongly imply that its cytoplasmic-nuclear translocation is a critical event regarding its role. However, this is hardly mentioned by the authors. Do they have any hint as to how is TDP43 translocated from the cytoplasm to the nucleus at the 2C stage? Has a functional role for this translocation of TDP-43v been described in other settings?
2. Related to the above, TDP-43 is located to the nucleus in the GV/FGO. Is it exerting any role here? Interestingly, at this stage there are more than 90 thousand specific binding peaks for TDP-43. Does this have any functional correlate? And is it nuclear right from the start?
3. The authors make quite a compelling case for TDP-43 regulating transcription through interaction with RNA pol II at the ZGA. However, is this a specific role only at this time window, or is it also occurring at other developmental stages or cell types? The authors have ES cells where they can conditionally remove TDP-43. What happens to these cells? Are there problems with localization of RNA pol II, or changes in gene expression related to TDP-43 binding? It would not be reasonable to ask the authors to answer these questions in the present manuscript, but at least this issue should be discussed (furthermore, as the Discussion is pretty short at present).
4. Does TDP-43 recruit Pol II, or is it only needed for its stabilization on chromatin? Who comes first? Is TDP43 bound correctly in the absence of Pol II (can this experiment be done at all)? It would also be extremely interesting to explore how TDP-43 dependant promoters behave along time, and if at some point they become independent. Could TDP-43 be involved in any way in polymerase pausing (or its release from pausing)? This is an issue that has not been explored or discussed at all in the manuscript.

SPECIFIC ISSUES

1. Why did the authors decide to explore the role of TDP-43 in ZGA? Some more background would be useful. What is the zygotic phenotype? This is hardly mentioned, and certainly of interest. This is even more confusing after reading all the information provided about its role in neurological pathologies, what is of little relevance to the results of the manuscript. There are more than 20 references for TDP-43 in the introduction. The authors should be more selective, choose the most relevant ones, and as said above, focus more on what is known about the zygotic mutant phenotype.

2. The authors must be careful with the use of abbreviations. These should be used only when really useful, say if they are used more than 4-5 times in the manuscript, or if they are obvious and well known (a good example of this would be ZGA). For example, OET (oocyte-to-embryo transition) is only used once (line 42), or on the other hand MZT (maternal-to-zygotic transition) is not defined (line 47). The authors should carefully check all of the text, as this occurs in other parts of the manuscript, as for example using NSN-FGO (line 218) only in this place instead of simply FGO, or using "zygote" in the text (line 215) or "1C" in Fig. 1a.

3. There are quite a number of typos, grammatical mistakes or missing words that should be checked throughout the manuscript, for example "...genetic disruption (of) most of these transcription factors..." (line 64), "...genes were involved (in) ribonucleoprotein..." (line 171), or "...we found (that) the change..." (line 249).

4. Regarding the dynamics of TDP-43 expression in the early embryo, the authors state that "...TDP-43 was still present in the cytoplasm of very early two-cell embryos" (lines 99-100). This seems to imply that this is the same pool of protein found in the cytoplasm of MII oocytes and 1-cell embryos. Is this the case? Do the authors have any evidence on maternal RNA?

5. The breeding scheme and description of the mice used should be much more detailed, in the main text and also in Methods. The generation of maternal mutants in mice using Zp3-Cre is not obvious, so this should be much better explained to readers, most of which will not be familiar with these genetic approaches. For example, which are the exact genotypes of the mice used to generate the Tdp43-oKO (Tdp43 flox/flox; Zp3-Cre). These do not come from crossing Tdp43 flox/flox mice with Zp3-Cre mice, as is stated in the text. In addition, that authors should state much more clearly in all described experiments which are the controls they are using.

6. In Supplementary Fig. 2, why is data shown in different ways in b, d and f? It would be best if all show each individual data point, as in b. The number of oocytes should be indicated in the figure legend for all graphs, here and elsewhere in other figure legends.

7. Is there any good reason for the authors to use the s.e.m. instead of the standard deviation (s.d.)? This last would be better to show the variability of the data.

8. As mentioned before, please specify in the legend to Fig. 2 the controls used for the RNA-seq. Also, only 2 replicates per stage and genotype were used. Wouldn't this be maybe too low? In any case the authors should explain and justify why 2 replicates are enough in their view.

9. The horizontal lines indicating the cut-offs for log₂ values in the volcano plot shown in Fig. 2b seem to be misplaced.

10. The quality of Supplementary Fig. 3 is very low, as text can hardly be read. This also happens in other figures.

11. TRIM-away is supposed to be very difficult or impossible for nuclear proteins, as you must get your antibody inside the nucleus. Do the authors think that in their case it worked ok because of the cytoplasmic localization of TDP-43 up to the early 2C-embryo (Fig. 1b)? If so, it would be good that the authors explained so, in order not to misguide other researchers with a false impression that the technique works ok for a nuclear factor.

12. For the Trim-away experiments shown in Fig. 2d-f and Supplementary Fig. 4b-f, it is necessary to provide more detail: how many embryos were used for RNA-seq in each condition? What explains the quite big differences in DEGs between KOs and KDs? How does the Trim-away RNA-seq data behave in a PCA when compared to the data shown in Fig. 2a?

13. The comparison of oKO and KD RNA-seqs shows a large proportion of DEGs specific to the Trim-away KDs. If the authors compare their standard control RNA-seq with the Trim-away IgG RNA-seq controls, do the same set of genes appear? Could these be considered as technique-specific changes? Again, this would be very useful for other researchers using r considering using Trim-away, and should be straightforward, as the RNA-seq data is already in place.

14. The authors state that the P-ser5 signal shown in Supplementary Fig. 5c was comparable between controls and mutants (lines 186-187). Why did they not quantify this signal, as was done for total Polr2a in Supplementary Fig. 5b?

15. Based on the immunos, maybe it is a bit too far to suggest that TDP-43 is involved in phase separation of Pol2 (line 192). Further evidence or explanations should be provided, or else the authors should remove this sentence.

16. Fig. 3 and Supplementary Fig. 5 are organized in a rather messy way, that implies the reader going back and forward to understand the results. The authors should make an effort to reorganize the panels to improve clarity. Also, and as noted above, the legends for Fig. 3 and Supplementary Fig. 5 should indicate exactly what is the control and number of embryos analysed in each case.

17. Regarding the reduced protein levels of Polr2a in TDP-43 maternal mutant embryos (Supplementary Fig. 5a, b), is their also a decrease in mRNA levels? In other words, does TDP-43 control the transcription of Pol2ra?

18. On the PLA experiments, there is a need for much more detail and a better description. A negative control should be included. At present, the authors use the KO embryos as controls. However, as these embryos do not express TDP-43, it is obvious that no PLA signal will be detected. It would be necessary to include an extra condition where control embryos are incubated with antibodies targeting TDP-43 and another protein, preferably nuclear, whose interaction with TDP-43 is not expected.

19. There is a need of a bit of introduction to Stacc-seq for non-experts in the main text, for example mentioning it is similar to others such as CUT&TAG, and ideal for samples with a limited number of cells. Also, the number of embryos used for each Stacc-seq replicate should be indicated; and are only two replicates enough?

20. Adding labels to Supplementary Fig. 7a-c would greatly help interpretation. This also applies to other figures in the manuscript, so it would be appreciated if the authors made the effort to revise them and add explanatory labelling where needed.

21. Was any statistical test run on the data shown in Supplementary Fig. 7a-c? And on that shown in Fig. 5b, d, e, or Fig. 6a, b? In general, statistical support for significant difference should be provided in all instances when the authors claim that there significant are differences between conditions.

22. The authors state that "...the genes bound more by TDP-43 at their promoters were more likely to be down-regulated upon the loss of TDP-43 (Fig. 6b, left panel). This was also true for major ZGA genes (Fig. 6b, right panel)" (lines 275-278). However, the data in Fig. 6b do not show so. First, if no statistical test is run (see above), the authors cannot claim any differences between quartiles that could seem to be present in the left panel. But, in the case of the right panel (ZGA genes), all quartiles look identical. Overall, this analysis is weak, and provides little support to the main conclusion of TDP-43 binding to ZGA genes.

23. Regarding the co-IPs shown in Fig. 6, why did the authors use a human cell line (HEK293T) transfected with overexpression vectors for TDP-43 and Ccnt1? Why no use mouse ES cells and endogenous proteins as before for TDP-43 and Pol2ra (Fig. 3c)? The antibodies for endogenous Ccnt1

seem to work ok for PLA. Furthermore, it would be good to have stainings showing the distribution of Ccnt1 in embryos, to see if they co-localize in foci with TDP-43 as Polr2a, and also to see if it is de-localized in the maternal Tdp43 mutant.

Reviewer #2 (Remarks to the Author):

The manuscript of Nie et al., investigates the process of zygotic genome activation (ZGA), which is of fundamental importance for the onset of embryonic development. Focusing on TDP-43 (TARDBP), the authors show that this factor is maternally provided and exhibits dynamic subcellular localisation. TDP-43 translocates to the nucleus at the early 2-cell stage, largely sharing genomic occupancy with Polr2a, promoting the expression of ZGA genes. Biochemical analyses indicate that TDP-43 forms a complex with Polr2a and Cyclin T1. Essentially, loss of function analysis revealed that TDP-43 depletion results in an early embryonic arrest, in agreement with its role in ZGA activation. This is a very well-crafted work, the manuscript is well-written, the data supports the main claims of the authors and the discoveries presented in this work are of great interest to the developmental biology field.

I have several questions and suggestions as follows.

1. Such nice work deserves more in-depth Discussion. It feels like the paper was intended for a journal such as Nature, where the word count is limited. I would suggest that the authors take advantage of the space provided by the Nature Communications format and extend the Discussion.
 2. In the Abstract the authors state that "early embryos arrested at late two-cell stage and female infertile". The statement is true, nevertheless, I would suggest not using infertile, as the authors nicely show that TDP-43 is not essential for fertilisation per se (text 128 – 135), so it might be a bit confusing for a non-expert.
 3. Some of the E1.5 Tdp43 oKO embryos in Figure 1C appear with an asymmetric size of the 2 blastomeres. Do these embryos exhibit blastomere fragmentation later?
 4. In figure 2C, the gene labels are too small.
 5. Can the authors provide a quantification of the co-localization of the TDP-43 and the p-Ser2 foci presented in Fig 3A. Are there pSer2 foci that do not co-localize with TDP-43? If yes, what could be the explanation?
 6. The authors state that "TDP-43 directly interacts with Pol II" (205 – 206). I would rather use "TDP-43 is in complex with Pol II". In principle, PLA shows that 2 factors are in close proximity, but not necessarily directly interacting. Co-IP experiments do not exclude the pull-down of a larger complex where two factors may not be necessarily directly bound.
 7. The pull-down on the right panel of Figure 3C (IP: Polr2a) shows TDP-43 enrichment in the IgG control. This has to be revised.
 8. It will be useful for the reader to mention briefly the principle of the Stacc-seq (line 210).
 9. Lines 282 -286. The authors state that "Pol II enters elongation for production when it is phosphorylated at its CTD Ser2 (P-Ser2) by P-TEFb (the positive transcription elongation factor complex, consisted of Cyclin T1/2 and CDK9, encoded by Ccnt1/2 and Cdk9)." Does Cdk9 exhibit similar subcellular distribution during ZGA as Cyclin T1? This can be discussed if the knowledge is available, as Cdk9 is not the focus of this study.
- Similarly, the Discussion can elaborate on the potential regulation of TDP-43 and Cyclin T1 translocation. It will be of interest to discuss/speculate on a potential interplay of other factors reported to be involved in ZGA and TDP-43/ Cyclin T1 described in this study, drawing a "big picture" of the ZGA.

Overall great study, keep up the good work!

We sincerely thank the editor and the reviewer for their valuable comments and suggestions, which have led to significant improvement of the manuscript. Here, we include a letter to address the issues raised in the previous submission. To avoid your confusion, we used Fig. 1-6 and Fig. S1-8 to refer the figures in the revised manuscript, and Reviewer only Figure I, II, III, etc. in this letter.

For the reviewers' convenience, we also included a version of the manuscript in which the revised key sections are highlighted. The POINT-BY-POINT RESPONSES to the reviewers' comments are as follows in bold font.

POINT-BY-POINT RESPONSES

Reviewer #1:

Response: We sincerely appreciate the reviewer for recognizing the merits of our work and offering constructive comments to enhance our manuscript. We have carefully considered the feedback and incorporated the suggestions into the revised manuscript. Detailed responses to each comment are provided below.

General Comment 1: The distribution of TDP-43 in the first stages of development, and the phenotypic consequence of the loss of maternal product, strongly imply that its cytoplasmic-nuclear translocation is a critical event regarding its role. However, this is hardly mentioned by the authors. Do they have any hint as to how is TDP43 translocated from the cytoplasm to the nucleus at the 2C stage? Has a functional role for this translocation of TDP-43 been described in other settings?

RESPONSES TO COMMENT 1:

We thank the reviewer for bringing up these important issues. We totally agree with your opinion that the translocation of TDP-43 from the cytoplasm to the nucleus is critical for its function. The mis-regulation of TDP-43 translocation is thought to be one of the concomitant events of cytoplasmic TDP-43 aggregation in amyotrophic lateral sclerosis (ALS) and frontotemporal lobar dementia (FTLD) patients¹. Thus, we have described TDP-43 cytoplasmic-nuclear translocation to strengthen the potential role in the revised manuscript (lines 114-116, lines 201-204, lines 288-290, lines 337-339, lines 363-368 and lines 373-389).

As described in the revised discussion (lines 376-386). TDP-43 contains

a classic NLS that have been shown to contacts importin a1/b and promotes TDP-43 translocation from the cytoplasm TDP-43 aggregation into the nucleus². The importin a (also known as karyopherin a, Kpna) family proteins include six members that highly express during mouse oocyte to embryo transition³ (Reviewer only Figure I). Maternal importin a2 (Kpna7) is required for mouse early embryonic development and fecundity⁴. Similar to those lacking maternal TDP-43, depletion of maternal importin a7 (Kpna6) results in the arrested embryo development at two-cell stage and the dramatically decreased expression of ZGA genes in mouse³. Thus, these importin proteins may interact with TDP-43 and regulate its translocation from the cytoplasm into the nucleus during mouse maternal to zygotic transition. Thus, the nuclear import adaptor proteins might be involved in TDP-43 translocation.

Reviewer only Figure I. The expression levels of *Kpna1*, *Kpna3*, *Kpna4*, *Kpna6* (*Importin a7*), *Kpna7* (*Importin a2*), *Kpnb1*, *Nup62* and *Tardbp* (TDP-43) in mouse preimplantation embryos detected by mass spectrometry⁵.

Yes, the mis-localization of TDP-43 from nucleus to the cytoplasm is usually observed in the cytoplasm aggregates in ALS and FTLN patients, presumably resulting in the nuclear function loss or cytoplasm toxicity in these diseases⁶. As described above, importin a1, KPNB1 and Nup62 was reported to resolve the cytoplasm aggregates of TDP-43 and rescue the hallmarks of TDP-43 pathology^{2, 7}. Thus, these observations suggest that TDP-43 translocation has the functional role in these cells.

General Comment 2: Related to the above, TDP-43 is located to the nucleus in the GV/FGO. Is it exerting any role here? Interestingly, at this stage there are more than 90 thousand specific binding peaks for TDP-43. Does this have any functional correlate? And is it nuclear right from the start?

RESPONSES TO COMMENT 2:

Thank you for your comments. These are quite important questions.

As described in the revised manuscript (lines 130-139), we did not find the obvious development phenotype in the oocytes with TDP-43 depletion by using ZP3-Cre, which depletes TDP-43 in the oocytes when they develop into the growth phase. Consistently, TDP-43 depletion barely affected the transcriptome in KO oocytes, as only 336 DEGs are observed in the KO oocytes (Supplementary Figure 3b). We further performed TDP-43 Stacc-seq in control and KO oocytes. Our results showed that TDP-43 binding preferentially at other genes, but not at the DEGs (different expression genes in *Tdp43^{oKO}* oocytes compared with the control) (Supplementary Fig. 8a). The distance from TSS to the nearest TDP-43 peaks was similar between the up- and down-regulated DEGs in full grown oocytes, further supporting that TDP-43 binding is not related with these gene expression (Supplementary Fig. 8b). We also performed the Stacc-seq for Pol II in the control and *Tdp43^{oKO}* oocytes. Our results showed that Pol II enrichment was almost not affected in *Tdp43^{oKO}* FGO compared with the control (Supplementary Fig. 8c,d), consistent with the observation that Pol II P-Ser2 foci were not affected in TDP43 KO oocytes (Supplementary Fig. 5g,h). The data of Supplementary Fig. 8a-d were described in the revised manuscript (lines 309-311). All together, these data suggest that the specific-binding peaks of TDP-43 is not related with the Pol II transcription and gene expression in full grown oocytes.

However, we also deleted maternal TDP-43 at an earlier timing of oogenesis with a *Vasa-Cre* mice in which the knockout efficiency of the target gene was reported to be more than 90% in mouse oocytes on E18.5⁸. After obtained *Tdp43^{fox/fox};Vasa-Cre* females, we examined the ovarian sections from these mice with HE staining. Our results showed that most of oocytes were loss in these samples (Figure II a). These results suggest that TDP-43 is critical for early process of mouse oogenesis. We think that the role and mechanism of TDP-43 in mouse oogenesis is deserved to be further investigated by an independent project.

Reviewer only Figure II, The representative ovary section from F/F (Tdp43^{Flox/Flox}, Control) and F/F;vasa-cre (Tdp43^{Flox/Flox};Vasa-Cre) females.

Yes, from our observations, TDP-43 is majorly in the nucleus of oocytes at early stage, probably similar to other cells in normal conditions⁹. In fact, we have stained the oocytes from D5, D10, D20 females. Our results show that the majority of TDP-43 was in the nucleus of mouse oocytes at all these stages. This data was added as Supplementary Fig. 1c and described the manuscript (lines 108-109), suggesting that TDP-43 starts to localize in the nucleus of mouse oocytes at least from primary follicle.

General Comment 3. The authors make quite a compelling case for TDP-43 regulating transcription through interaction with RNA pol II at the ZGA. However, is this a specific role only at this time window, or is it also occurring at other developmental stages or cell types? The authors have ES cells where they can conditionally remove TDP-43. What happens to these cells? Are there problems with localization of RNA pol II, or changes in gene expression related to TDP-43 binding? It would not be reasonable to ask the authors to answer these questions in the present manuscript, but at least this issue should be discussed (furthermore, as the Discussion is pretty short at present).

RESPONSES TO COMMENT 3:

We thank the reviewer very much for appreciating the merits of the function of TDP-43 on Pol II and the comments.

As shown in this study, TDP-43 functions together with cyclinT1 to

regulate Pol II elongation for the expression of ZGA genes in mouse 2-cell embryos. However, as described in RESPONSES TO COMMENT 2, Pol II distribution was not significantly affected in *Tdp43^{oKO}* full grown oocytes (Supplementary Fig. 8c,d), and the DEGs were not related to the binding of TDP-43 in mouse oocytes (Supplementary Fig. 8a,b). Thus, these observations suggest that TDP-43 regulating transcription through RNA Pol II is a stage specific role during mouse oocyte to embryo transition.

Actually, we mated *Tdp43^{flox/Flox} mcie* with *ERT2-Cre mice* to establish a Tamoxifen-Cre(ERT-cre) induced TDP-43 KO ESCs. After the treatment of Tamoxifen for 72 hours, TDP-43 was dramatically decreased in these cells, thus as TDP-43 induced knock-out mESCs (iKO ESCs) (Reviewer only Figure III a,b). Compared with the controls (Ctl), the TDP-43 iKO ESCs expressed similar levels of pluripotency markers (Reviewer only Figure III c, d). However, TDP-43 iKO ESCs could not be passaged and lose their potential of self-renewal (data not shown).

We performed immunofluorescence staining with P-Ser2 antibody for TDP-43 iKO ESCs. The pattern of Pol II P-Ser2 was similar in both iKO and Ctl ESCs (Figure III e), similar to that in TDP-43 KO oocytes (Supplementary Fig. 5g,h). We also performed RNA-seq for these iKO ESCs. Surprisingly, our results showed that TDP-43 iKO ESCs had very limited DEGs (359 upregulated and 288 down-regulated) when compared with the control (Reviewer only Figure III f). In addition, by performing Pol II Stacc-seq, we found that the correlation of Pol II enrichment was very high between Ctrl and iKO ESCs (Pearson's correlation coefficients >0.94), suggesting that the Pol II distribution is almost not affected after the deletion of TDP-43 in mouse ESCs (Reviewer only Figure III g, h). To investigate whether these DEGs was related to TDP-43 binding, we tried to performed Stacc-seq with the TDP-43 antibody for mouse ESCs. We also tried to performed the conventional Chip-seq with Flag antibody for the ESCs with the overexpression of Flag-TDP-43. However, both antibodies didn't work well in mouse ESCs. We are still working on these issues.

A previous study showed that both TDP-43 and Pol II binds the promoter of *Acrv1* in liver cell and spermatocytes and round spermatids. However, the expression of *Acrv1* is only in round spermatids¹⁰. These authors proposed that TDP-43 is a transcriptional repressor for the testis-specific mouse *acrv1* gene⁹.

Taken together, these data suggest that TDP-43 regulates Pol II transcription in a specific- or context-dependent manner. As you

suggested, we have discussed these context-specific regulation of TDP-43 on the active transcription of Pol II in the revised manuscript (lines 391-402).

[redacted]

General Comment 4. Does TDP-43 recruit Pol II, or is it only needed for its stabilization on chromatin? Who comes first? Is TDP43 bound correctly in the absence of Pol II (can this experiment be done at all)? It would also be extremely interesting to explore how TDP-43 dependent promoters behave along time, and if at some point they become independent. Could TDP-43 be involved in any way in polymerase pausing (or its release from pausing)? This is an issue that has not been explored or discussed at all in the manuscript.

RESPONSES TO COMMENT 4:

These are very important questions to address how TDP-43 function in mouse ZGA. However, some experiments are hard to perform in mouse early embryos.

Currently, we do not think that TDP-43 directly recruits Pol II to ZGA genes. The translocation of TDP-43 from the cytoplasm into the nucleus is in the company of Pol II pre-configuration during mouse ZGA. Thus, the simultaneous appearance of both Pol II and TDP-43 at ZGA genes is possible. On the other hand, in TDP-43 KO L2C embryos, no significant change of Pol II enrichment was observed at the promoters of major ZGA genes (Fig. 5d,e,f), suggesting TDP-43 plays mild role in the Pol II binding at their promoters.

We do not know whether TDP-43 is exactly needed for Pol II stabilization on chromatin. Because the lost of TDP-43 led to little alteration on Pol II configuration at L2C specific targets (Fig. 5a), TDP-43 may not be needed for Pol II stabilization on chromatin.

As described above, simultaneous appearance of Pol II and TDP-43 at ZGA genes is possible.

To investigate whether TDP43 bounds correctly in the absence of Pol II, mouse 2-cell embryos were treated with DRB for 7 hours. However, we found that both Pol II and TDP-43 were dramatically decreased in 2-cell embryos (Reviewer only Figure IV a). Previous study showed that Pol II was degraded by α -amanitin treatment in mouse embryos because this treatment induced dramatically apoptosis and led to a global protein degradation^{11, 12}. Therefore, DRB treatment might also induced the

apoptosis and resulted in a global protein degradation including TDP-43. Thus, TDP43 binding in the absence of Pol II is technically difficult.

We totally agree with that it will be great interesting to explore how TDP-43 dependent promoters behave during mouse pre-implantation development. However, this idea is very difficult to be directly tested. Using mouse embryos for the Stacc-seq experiments is very timing consumption and expensive cost. Alternatively, to explore how TDP-43 dependent promoters behave during the development, we identified the expressed genes with TDP-43 binding at their promoter in late two-cell embryos. Then, we examined the expression of these genes in 4-cell, morula and blastocyst stage of normal embryos with the published data¹³. We found that the expression of these genes starts to increase at late 2-cell embryos, then maintained a relatively high expression level until to 8-cell stage, while started decrease after 8-cell embryos stage. These data was added as Supplementary Fig. 7g and described in the revised manuscript (lines 307-308). These data suggest that the promoters of these genes are active in mouse embryos from late two cell until 8-cell stage.

Yes, our results suggest that TDP-43 recruits Cyclin T1 to phosphorylates Pol II and promotes Pol II to release from pausing to initiate elongation. To explore how TDP-43 function in Pol II, we made great efforts on the relationship between TDP-43 and Cyclin T1, a component of P-TEFb (the positive transcription elongation factor complex. Firstly, we performed Co-IP and found that TDP-43 interacted with Cyclin T1 in 293 cells (Fig. 6d,e). We also performed PLA experiments and found the interaction between TDP43 and Cyclin T1 in mouse two-cell embryos (Fig. 6f, g, Supplementary Fig. 8f).

Moreover, we tried to over-express TDP-43-NLS and Cyclin T1 in mouse 1-cell embryos which have TDP-43 and Cyclin T1 in their cytoplasm (Reviewer only Figure IV b). Strikingly, the enforced TDP-43 into the nucleus of 1-cell embryos recruited Cyclin T1 into the nucleus, but the nuclear Cyclin T1 could not recruit TDP-43 into the nucleus (Reviewer only Figure IV c). We were so pleased to repeat this experiment (Reviewer only Figure IV d). However, we run out the antibody of Cyclin T1 (Abcam 184703, lot GR250586-7) when we tried to repeat these experiments again. We had ordered two new antibodies (Abcam 184703, lot 1011450-2; Abcam cat 184703, lot GR250586-10) for Cyclin T1 staining. Unfortunately, we found that cyclin T1 was already in the nucleus of 1-cell embryos (data not shown). The nuclear location of Cyclin T1 in mouse 1-cell embryos is not consistent with previous report and our published results^{11,18}. These results very confused us and we could not

repeat the translocation of TDP-43 in mouse zygotes due to the antibody. Thus, we would not like to present these results in the revised manuscript. Anyway, these observations suggest that TDP-43 translocation into the nucleus can recruit the Cyclin T1 into the nucleus in mouse 1-cell embryos (Reviewer only Figure IV b,c,d), further supporting that TDP-43 interacts Cyclin T1 to phosphorylate Pol II and assists Pol II release from its pausing during mouse ZGA. We have added this information in the revised discussion (lines 366-368).

Figure IV for Reviewer only:

Reviewer only Figure IV a, The staining of TDP-43 and Pol II in the embryos treated with DMSO or DRB for 7 hours. Scale bar, 10 μ m. **b**, The staining of TDP-43 and Cyclin T1 in wild type 1-cell embryos. **c,d**, The staining of endogenous of TDP-43 or Cyclin T1 in 1-cell embryos with the overexpression of Ccnt1-EGFP and NLS-TDP-43-mcherry, scale bar, 10 μ m. **e**, The EGFP-CCnt1 in Tdp43^{oKO} L2C embryos that were injected with EGFP-Ccnt1 mRNA at 1-cell stage. Scale bar, 10 μ m.

SPECIFIC ISSUES

Specific Comment 1: Why did the authors decide to explore the role of TDP-43 in ZGA? Some more background would be useful. What is the zygotic phenotype? This is hardly mentioned, and certainly of interest. This is even more confusing after reading all the information provided about its role in neurological pathologies, what is of little relevance to the results of the manuscript. There are more than 20 references for TDP-43 in the introduction. The authors should be more selective, choose the most relevant ones, and as said above, focus more on what is known about the zygotic mutant phenotype.

RESPONSES: Thanks for raising these questions. From the background of current knowledge, it is very difficult to relate TDP-43 to mouse ZGA. Thus, we have added some information about ZGAIN in the revised introduction (lines 42-43, lines 85-87).

Due to the early embryonic lethality during peri-implantation, three independent groups reported the very limited information for TDP-43 zygotic phenotype^{14, 15, 16}. We have now added more information in the revised manuscript (lines 80-85).

We have now removed most of the descriptions and the references about neurological pathologies in the revised manuscript to avoid the confusion (lines 68-77).

Specific Comment 2: The authors must be careful with the use of abbreviations. These should be used only when really useful, say if they are used more than 4-5 times in the manuscript, or if they are obvious and well know (a good example of this would be ZGA). For example, OET (oocyte-to-embryo transition) is only used once (line 42), or on the other hand MZT (maternal-to-zygotic transition) is not defined (line 47). The authors should carefully check all of the text, as this occurs in other parts of the manuscript, as for example using NSN-FGO (line 218) only in this place instead of simply FGO, or using “zygote” in the text (line 215) or “1C” in Fig. 1a.
RESPONSES: We appreciate you for pointing out the issue. We have paid more attention to the abbreviations in the revised manuscript.

Specific Comment 3. There are quite a number of typos, grammatical mistakes or missing words that should be checked throughout the manuscript, for example “...genetic disruption (of) most of these transcription factors...” (line 64), “...genes were involved (in) ribonucleoprotein...” (line171), or “...we found (that) the change...” (line 249).

RESPONSES: Thank you for your corrections. We have now checked throughout the manuscript and revised the typos in the revised manuscript.

Specific Comment 4. Regarding the dynamics of TDP-43 expression in the early embryo, the authors state that “...TDP-43 was still present in the cytoplasm of very early two-cell embryos” (lines 99-100). These seems to imply that this is the same pool of protein found in the cytoplasm of MII oocytes and 1-cell embryos. Is this the case? Do the authors have any evidence on maternal RNA?

RESPONSES: We thank the reviewer for pointing out this interesting observation. Yes, our results supported your speculation. In fact, we had

checked our published data of Rio-seq (translation) and mRNA-seq (transcription)¹⁷. We found that the expression of TDP-43 continuously decreases after fertilization until late 2-cell at both transcription and translation level (Supplementary Fig. 1a), suggesting that maternal TDP-43 regulates the ZGA in pre-implantation embryos (at least in two-cell embryos). We had also examined the expression level of TDP-43 protein by Western blot and found a similar trends to the mRNAs (Supplementary Fig. 1b). We have now added this information in the revised manuscript (lines 99-106).

Specific Comment 5. The breeding scheme and description of the mice used should be much more detailed, in the main text and also in Methods. The generation of maternal mutants in mice using Zp3-Cre is not obvious, so this should be much better explained to readers, most of which will not be familiar with these genetic approaches. For example, which are the exact genotypes of the mice used to generate the Tdp43-oKO (Tdp43^{flox/flox};Zp3-Cre). These do not come from crossing Tdp43 flox/flox mice with Zp3-Cre mice, as is stated in the text. In addition, that authors should state much more clearly in all described experiments which are the controls they are using.

RESPONSES: Thanks for the suggestions that can make our study to be more widely understood. We have amended the description of breeding strategy in the main text (lines 118-121) and in Material and Method (lines 411-416).

Specific Comment 6: In Supplementary Fig. 2, why is data shown in different ways in b, d and f? It would be best if all show each individual data point, as in b. The number of oocytes should be indicated in the figure legend for all graphs, here and elsewhere in other figure legends.

RESPONSES: We thank the reviewer for these suggestions. We have now added individual data in the Supplementary Fig. 2 d,f, and added the number of oocytes in figure legend of Supplementary Fig. 2. Similar to this treatment, Other figures and their legends have be revised (Fig.3&6 Supplementary Fig.5).

Specific Comment 7: Is there any good reason for the authors to use the s.e.m. instead of the standard deviation (s.d.)? This last would be better to show the variability of the data.

RESPONSES: Thank you for your suggestions. There is no specific reason for using s.e.m. instead of s.d. just because we usually used the s.e.m for the statistical analysis in the laboratory. Of course, we agree with that s.d. is better to show the variability of data. Thus, we have now revised all of the quantitative analysis with s.d. for standard deviation in the revised manuscript.

Specific Comment 8: As mentioned before, please specify in the legend to Fig. 2 the controls used for the RNA-seq. Also, only 2 replicates per stage and genotype were used. Wouldn't this be maybe too low? In any case the authors should explain and justify why 2 replicates are enough in their view.

RESPONSES: We appreciate the reviewer to point this out. We have now specified our control in the legend of Fig. 2.

Given the limited samples of mammalian oocytes and embryos, we have developed a stable and reproducible technique for RNA-seq during the past several years^{11, 13}. Thus, we generated the replicates of RNA-seq for each strategy (*Tdp43*^{oKO} or Trim-away TDP-43 KD samples) in two independent experiments. The replicates were highly reproducible ($R > 0.95$, Supplementary Fig. 3a). As your suggested, we have added the justification when we used in the study.

Specific Comment 9: The horizontal lines indicating the cut-offs for log2 values in the volcano plot shown in Fig. 2b seem to be misplaced.

RESPONSES: Thank you for your correction, we have corrected the mistake in the Fig. 2b.

Specific Comment 10: The quality of Supplementary Fig. 3 is very low, as text can hardly be read. This also happens in other figures.

RESPONSES: Thanks for pointing this out. We have promoted the quality of Supplementary Fig. 3. We have also revised other figures, such Supplementary Fig. 4.

Specific Comment 11: TRIM-away is supposed to be very difficult or impossible for nuclear proteins, as you must get your antibody inside the nucleus. Do the authors think that in their case it worked ok because of the cytoplasmic localization of TDP-43 up to the early 2C-embryo (Fig. 1b)? If so, it would be good that the authors explained so, in order not to misguide other researchers with a false impression that the technique works ok for a nuclear factor.

RESPONSES: We agree with that the efficiency of TDP-43 knocked-down by Trim-away in this study might rely on the cytoplasmic localization of TDP-43 up to mouse early 2C-embryo. We have now added this information in the revised manuscript (lines 181-182).

Specific Comment 12: For the Trim-away experiments shown in Fig. 2d-f and Supplementary Fig. 4b-f, it is necessary to provide more detail: how many embryos were used for RNA-seq in each condition? What explains the quite big differences in DEGs between KOs and KDs? How does the Trim-away RNA-seq data behave in a PCA when compared to the data shown in Fig. 2a?

RESPONSES: We thank the reviewer for pointing this out. We used

twenty embryos for RNA-seq with embryos injected *Trim21* mRNA and the antibody against TDP-43 or rabbit IgG, and we have added this information in the revised manuscript (lines 182-183).

For the different DEGs identified in KO/KD embryos generated by two different strategies for the depletion of TDP-43, we think that the different DEGs might potentially come from following sources: 1) maternal depleted TDP-43 KO embryos have potential defects that could bring from the oocytes, which might persist to the embryo stage and cause the secondary defects; 2) TDP-43 KO embryos generated by Trim-away are supposed to have less indirect defects on the transcriptome of 2-cell embryos. However, the quality of the data from Trim-away is very dependent on the conditions of in vitro culture and is also very dependent on the antibody specification. 3) KO embryos were isolated from *Tdp43*^{oKO} females with a mixed background of C57BL/6J and ICR, while KD embryos were isolated from wild-type commercial ICR females (Materials and Methods). The different backgrounds probably gave rise to the DEGs. 4) The different development time of the embryos used in the two strategies was not synchronous (Revised Supplementary Fig. 4d), which could be another reason. Altogether, these are the technique challenges for these experiments.

We have performed PCA analysis and found KD embryos were more closed to the counterpart control *in vivo*. We have now added these data as Supplementary Fig. 4 d and described this information in the revised manuscript (lines 196-197).

Specific Comment 13: The comparison of oKO and KD RNA-seqs shows a large proportion of DEGs specific to the Trim-away KDs, with . If the authors compare their standard control RNA-seq with the Trim-away IgG RNA-seq controls, do the same set of genes appear? Could these be considered as technique-specific changes? Again, this would be very useful for other researchers using or considering using Trim-away, and should be straightforward, as the RNA-seq data is already in place.

RESPONSES: We thank the reviewer for the suggestion. As suggested, we compared the RNA-seq data of control for oKO and Inj-control for KD RNA-seq. Our results showed that there were many DEGs between these two controls. However, limited genes were overlapped between these experiments (Reviewer only Figure VIII). These data further support that “technic-specific changes” in these experiments, probably caused by the reasons as described in the RESPONSES to comment 12. In our experience, the critical controls and careful confirmations are very necessary for the using of these techniques. Thus, we set two

independent control for each technique.

Reviewer only Figure V: **a**, Volcano plots showing the different expression genes (DEGs) between the control of Tdp43^{KO} and the Inj-Control (IgG) of KD (adjust P-value, 0.05; fold-change, 2). **b,c,d** the Venn diagram showing the number of overlapped genes with the down- and up-regulated or total DEGs among Tdp43^{KO}/ KD L2C embryos and two controls.

Specific Comment 14: The authors state that the P-ser5 signal shown in Supplementary Fig. 5c was comparable between controls and mutants (lines 186-187). Why did they not quantify this signal, as was done for total Polr2a in Supplementary Fig. 5b?

RESPONSES: Thanks for pointing this out. As suggested, we have now quantified the signal intensity of P-ser5, the result show there was no significant different between the control and Tdp43^{KO}. And we added these data as Supplementary Fig. 5e.

Specific Comment 15: Based on the immunos, maybe it is a bit too far to suggest that TDP-43 is involved in phase separation of Pol2 (line 192). Further evidence or explanations should be provided, or else the authors should remove this sentence.

RESPONSES: As suggested, we have removed this statement from the revised manuscript.

Specific Comment 16: Fig. 3 and Supplementary Fig. 5 are organized in a rather messy way, that implies the reader going back and forward to understand the results. The authors should make an effort to reorganize the panels to improve clarity. Also, and as noted above, the legends for Fig. 3 and Supplementary Fig. 5 should indicate exactly what is the control and number of embryos analysed in each case.

RESPONSES: We appreciate the reviewer for pointing these issues out, To make the figures more understandable to the readers, we have reorganized the panels for the Fig. 3 and Supplementary Fig. 5 as the following: 1): we have removed the PLA channel of original Fig. 3e, and integrated the merged channel of PLA and DAPI of original Fig. 3e

(control) and the original Supplementary Fig. 5g (Tdp43oKO) into the new Fig. 3e; 2): a similar integration for PLA results of P-ser2 and TDP-43 (original Supplementary Fig. 5h,i) as the revised Supplementary Fig. 5k.

We have also added number of the embryos analyzed in their legends for these figures in the revised manuscript.

Specific Comment 17: Regarding the reduced protein levels of Polr2a in TDP-43 maternal mutant embryos (Supplementary Fig. 5a, b), is their also a decrease in mRNA levels? In other words, does TDP-43 control the transcription of Pol2ra?

RESPONSES: From the data of RNA -seq and qPCR, Polr2a mRNA showed no significant change between Tdp43^{oKO} and control embryos (Supplementary Fig. 5c,d). Thus, these results suggest that TDP-43 could not control the transcription of Pol2ra in mouse two-cell embryos. We have added these data as the revised Supplementary Fig. 5c,d and described this information in the revised manuscript (lines 206-207)

Specific Comment 18: On the PLA experiments, there is a need for much more detail and a better description. A negative control should be included. At present, the authors use the KO embryos as controls. However, as these embryos do not express TDP-43, it is obvious that no PLA signal will be detected. It would be necessary to include an extra condition where control embryos are incubated with antibodies targeting TDP-43 and another protein, preferably nuclear, whose interaction with TDP-43 is not expected.

RESPONSES: As suggested, we performed the PLA of TDP-43 and Tead1, a transcription factors as additional negative control. These data have been added as the revised Supplementary Fig. 5j and described in the revised manuscript (lines 226-228).

Specific Comment 19: There is a need of a bit of introduction to Stacc-seq for non-experts in the main text, for example mentioning it is similar to others such as CUT&TAG, and ideal for samples with a limited number of cells. Also, the number of embryos used for each Stacc-seq replicate should be indicated; and are only two replicates enough?

RESPONSES: As suggested, we have added more description of the Stacc-seq in the revised manuscript (lines 237-238).

We performed the Stacc-seq with at least 150 embryos or oocytes for each experiment of each antibody against TDP-43 (~200) or Pol II (~150), we have added this information in the materials and methods (lines 563-564).

Because of the limitation of mammalian early embryos and oocytes, we

have developed a sensitive and reliable Stacc-seq for these samples¹¹. Of course, the quality and reliability of Stacc-seq are very dependent on the antibody and the samples. Generally, we perform two replicates with two independent experiments with critical validation for these experiments (Supplementary Fig. 5a-e). After carefully checking the quality and reliability of Stacc-seq data, we will use these data for the next study. Otherwise, we will repeat the experiments depending on the specification of antibody and samples. In this study, our results were high reproducible and reliable (Supplementary Fig. 5a-e). Thus, we think that two replicates are enough to make the observation and conclusion in this study.

Specific Comment 20: Adding labels to Supplementary Fig. 7a-c would greatly help interpretation. This also applies to other figures in the manuscript, so it would be appreciated if the authors made the effort to revise them and add explanatory labelling where needed.

RESPONSES: As suggested, we have now added the labels for Supplementary Fig. 7a-c, and have tried to add explanatory labels that we think where are necessary.

Specific Comment 21: Was any statistical test run on the data shown in Supplementary Fig. 7a-c? And on that shown in Fig. 5b, d, e, or Fig. 6a, b? In general, statistical support for significant difference should be provided in all instances when the authors claim that there significant are differences between conditions.

RESPONSES: Thanks for the suggestions. We performed statistical test with paired two-side Wilcoxon rank-sum to investigate whether there is significant difference between these conditions. We have added the information in the revised Supplementary Fig. 7a-c, and Fig. 5b, d, e, or Fig. 6a, b. We also carefully checked the manuscript when we compared the difference between conditions.

Specific Comment 22: The authors state that "...the genes bound more by TDP-43 at their promoters were more likely to be down-regulated upon the loss of TDP-43 (Fig. 6b, left panel). This was also true for major ZGA genes (Fig. 6b, right panel)" (lines 275-278). However, the data in Fig. 6b do not show so. First, if no statistical test is run (see above), the authors cannot claim any differences between quartiles that could seem to be present in the left panel. But, in the case of the right panel (ZGA genes), all quartiles look identical. Overall, this analysis is weak, and provides little support to the main conclusion of TDP-43 binding to ZGA genes.

RESPONSES: We appreciate you for pointing these out. We have now reanalyzed the data with statistical test. Our results show that

expression of the genes bound by TDP-43 (for all genes) among the Q1, Q2, Q3, Q4 were significant difference upon the loss of TDP-43, although the expression of ZGA genes bound by TDP-43 was less significance of the difference upon the loss of TDP-43 according to the same categories (Fig. 6b, Reviewer only Figure VI a). However, when these genes were separated in bound or unbound by TDP-43, these differences are more obvious (Fig. 6c, Reviewer only Figure VI b). We think that these observations are interesting and have described a mild relation between the gene expression and TDP-43 binding. To avoid the overstatement, we have now deleted “This was also true for major ZGA genes” in the revised manuscript. Taking together, we would like to keep these figures and the statements in the revised manuscript (lines 306-308). Thank you again for your suggestions.

Reviewer only Figure VI a. The box plots showing the mRNA fold-changes (log 2 ratio) between *Tdp43*^{oKO} (oKO) and control in L2C for all genes (left) and major ZGA genes (right) covered or uncovered by TDP-43 at the promoters. The promoters unbound by TDP-43 were characterized as Supplementary Fig. 7f. **b** The box plots showing the mRNA fold-changes (log 2 ratio) between *Tdp43*^{oKO} (oKO) and control in L2C for all genes (left) and major ZGA genes (right) covered or uncovered by TDP-43 at the promoters. Upper and lower line of the box representing 25th and 75th percentiles. Whiskers showing 1.5X interquartile range. Central line, the median.

Specific Comment 23: Regarding the co-IPs shown in Fig. 6, why did the authors use a human cell line (HEK293T) transfected with overexpression vectors for TDP-43 and *Ccnt1*? Why no use mouse ES cells and endogenous proteins as before for TDP-43 and *Pol2ra* (Fig. 3c)? The antibodies for endogenous *Ccnt1* seem to work ok for PLA. Furthermore, it would be good to have stainings showing the distribution of *Ccnt1* in embryos, to see if they co-localize in foci with TDP-43 as *Pol2ra*, and also to see if it is de-localized in the maternal *Tdp43* mutant.

RESPONSES:

Since the Ccnt1 antibody did not work well for Co-IP experiment in mouse ESCs, we performed these experiments with the overexpression vectors in HEK293T cells for the interaction between TDP-43 and Ccnt1 (Fig. 6d,e).

The Ccnt1 used for PLA is suitable for the staining in mouse early embryos (Figure IVb). However, as described above, this antibody (Abcam 184703, lot GR250586-7) was run out. Then, we bought another two batch of Abcam antibody. We stained mouse embryos with the new antibody, there was obvious signal in the nucleus of 1-cell embryos. These data were very different from the results in previous report and our previous results^{11, 18}, making us very confuse. Thus, we decided to investigate whether Ccnt1 co-localizes with TDP-43 in foci of P-ser2 in mouse 1-cell embryos with the injection of mRNA of Ccnt1-EGFP. After injection, we cultured the embryos and observed the distribution of them in 2-cell embryos with immunostaining. Our results show that the injected Ccnt1 partly colocalized with TDP-43 in the foci of TDP-43 and P-ser2 in the nuclei of 2-cell embryos. We have added these data as revised Supplementary Fig. 8f and described in the revised manuscript (lines 330-331).

The injected Ccnt1 was diffused in the cytoplasm of the maternal Tdp43 mutant embryos (Reviewer only Figure IV 4e).

Reviewer #2 (Remarks to the Author):

The manuscript of Nie et al., investigates the process of zygotic genome activation (ZGA), which is of fundamental importance for the onset of embryonic development. Focusing on TDP-43 (TARDBP), the authors show that this factor is maternally provided and exhibits dynamic subcellular localisation. TDP-43 translocates to the nucleus at the early 2-cell stage, largely sharing genomic occupancy with Polr2a, promoting the expression of ZGA genes. Biochemical analyses indicate that TDP-43 forms a complex with Polr2a and Cyclin T1. Essentially, loss of function analysis revealed that TDP-43 depletion results in an early embryonic arrest, in agreement with its role in ZGA activation.

This is a very well-crafted work, the manuscript is well-written, the data supports the main claims of the authors and the discoveries presented in this work are of great interest to the developmental biology field.

Response: We thank the reviewer very much for appreciating the merits of this work and for providing constructive comments to strengthen our manuscript. We have followed the advice and revised the manuscript. The point-by-point responses are listed below.

I have several questions and suggestions as follows.

Specific Comment 1: Such nice work deserves more in-depth Discussion. It feels like the paper was intended for a journal such as Nature, where the word count is limited. I would suggest that the authors take advantage of the space provided by the Nature Communications format and extend the Discussion.

RESPONSES TO COMMENT 1:

We thank the reviewer for these comments and suggestions. As you suggested, we have revised and extended the discussion in the revised manuscript (lines 336-404).

Specific Comment 2: In the Abstract the authors state that “early embryos arrested at late two-cell stage and female infertile”. The statement is true, nevertheless, I would suggest not using infertile, as the authors nicely show that TDP-43 is not essential for fertilization per se (text 128 – 135), so it might be a bit confusing for a non-expert.

RESPONSES TO COMMENT 2:

As suggested, we have removed “and female infertile” in the Abstract of revised manuscript (lines 26-27).

Specific Comment 3: Some of the E1.5 Tdp43^{oKO} embryos in Figure 1C appear with an asymmetric size of the 2 blastomeres. Do these embryos

exhibit blastomere fragmentation later?

RESPONSES TO COMMENT 3:

Thank you for your comments. We separated the symmetric and asymmetric embryos that isolated from Tdp-43 KO females and cultured these embryos for additional 3 days. Many of these embryos exhibited blastomere fragmentation or die (Figure VI). We have also added some information into the revised manuscript (line 146).

Reviewer only Figure VII a, 2-cell embryos were collected from control and *Tdp43*^{oKO} mice, the symmetric and asymmetric *Tdp43*^{oKO} embryos were separated, all 2-cell embryos were culture for additional 3 days, and they were imaged with Zeiss Airyscan880. **b**, The rate of fragment or die embryos (as shown in Figure VI, a) in symmetric and asymmetric *Tdp43*^{oKO} group at E4.5, the embryos were separated to symmetric or asymmetric Tdp43oKO group when they were 2-cell embryos.

Specific Comment 4: In figure 2C, the gene labels are too small.

RESPONSES TO COMMENT 4:

Thank you for your comments. We have enlarged the gene labels in Fig. 2c in the revised manuscript.

Specific Comment 5: Can the authors provide a quantification of the co-localization of the TDP-43 and the p-Ser2 foci presented in Fig 3A. Are there pSer2 foci that do not co-localize with TDP-43? If yes, what could be the explanation?

RESPONSES TO COMMENT 5:

We appreciate reviewer for these comments. It's very hard to quantify these dispersed signals in Fig. 3a. Alternatively, to quantify the co-localization of the TDP-43 and the p-Ser2 foci, we simultaneously treated the embryos with 4% PFA and 0.5% Trinton X-100 for 25 mins as

in Fig 3b. We carefully investigated the co-localization of the TDP-43 and the p-Ser2 foci in ten late 2-cell embryos throughout the nucleus with every micro by LSM880 microscope. All of the P-Ser2 foci were co-localized with those of TDP-43 as shown in the following as a example. We described this information in the revised manuscript (lines 214-215).

Reviewer only Figure VIII Immunostaining of P-Ser2 and TDP-43 in normal L2C. Normal two-cell embryos were simultaneously treated with 4% PFA and 0.5% Triton X-100 for 25 mins and fixed. These embryos were stained with P-Ser2 and TDP-43 antibody. The pictures were obtained with LSM880 microscope.

Specific Comment 6: The authors state that “TDP-43 directly interacts with Pol II” (205 – 206). I would rather use “TDP-43 is in complex with Pol II”. In principle, PLA shows that 2 factors are in close proximity, but not necessarily directly interacting. Co-IP experiments do not exclude the pull-down of a larger complex where two factors may not be necessarily directly bound.

RESPONSES TO COMMENT 6:

As suggested, we have now removed “directly” in the revised manuscript.

Specific Comment 7: The pull-down on the right panel of Figure 3C (IP: Polr2a) shows TDP-43 enrichment in the IgG control. This has to be revised.

RESPONSES TO COMMENT 7:

We appreciate reviewer for pointing this out. The molecular weight of TDP-43 (43kDa) is closed to that of IgG heavy chain of (about 55kDa) in 12% Western bot gel. so some times when PVDF membrane was over exposing, the heavy chain could cover the position of 43KD, so it seems there is signal in the IgG line. We have repeated the Co-IP, and the Co-IP samples were run with the low concentration (10%) of Western bot gel to separate the 36-55Da. We obtained much better results for this experiment. The result was used to replace the original as shown in the revised Fig. 3c. Thank you again for this comment.

Specific Comment 8: It will be useful for the reader to mention briefly the principle of the Stacc-seq (line 210).

RESPONSES TO COMMENT 8:

We thank for the suggestion. We have briefly descried the principle of the Stacc-seq in the revised manuscript (lines 237-238).

Specific Comment 9: Lines 282 -286. The authors state that “Pol II enters elongation for production when it is phosphorylated at its CTD Ser2 (P-Ser2) by P-TEFb (the positive transcription elongation factor complex, consisted of Cyclin T1/2 and CDK9, encoded by Ccnt1/2 and Cdk9).” Does Cdk9 exhibit similar subcellular distribution during ZGA as Cyclin T1? This can be discussed if the knowledge is available, as Cdk9 is not the focus of this study.

RESPONSES TO COMMENT 9:

We thank the reviewer for the suggestions. We have performed the staining of CDK9 with three antibodies. Two antibodies from Proteintech (11705-1-AP) and Abclonal (A0886) did not work for staining. The results from the third antibody of Abcam (ab76320) showed signal in the nucleus at both 1-cell and 2-cell stage, which shows different expression dynamics as cyclinT1 during ZGA. Because Cyclin T2 had be reported in the pronucleus of mouse zygotes¹⁹, these results indicate that CDK9 might be a partner with Cyclin T2 to initiate minor ZGA in mice. We have now added this information as the revised Supplementary Fig. 8e and described them in the revised manuscript (lines 322-323).

Similarly, the Discussion can elaborate on the potential regulation of TDP-43 and Cyclin T1 translocation. It will be of interest to discuss/speculate on a potential interplay of other factors reported to be involved in ZGA and TDP-43/ Cyclin T1 described in this study, drawing a “big picture” of the ZGA.

Overall great study, keep up the good work!

RESPONSES: These are very good suggestions and gave great

encouragement for us. Limited by the available data, we have discussed the potential regulation of TDP-43 and Cyclin T1 translocation in the revised manuscript (lines 363-368 and lines 373-389).

1. Prasad A, Bharathi V, Sivalingam V, Girdhar A, Patel BK. Molecular Mechanisms of TDP-43 Misfolding and Pathology in Amyotrophic Lateral Sclerosis. *Front Mol Neurosci* **12**, (2019).
2. Doll SG, *et al.* Recognition of the TDP-43 nuclear localization signal by importin alpha1/beta. *Cell reports* **39**, 111007 (2022).
3. Rother F, *et al.* Importin alpha7 is essential for zygotic genome activation and early mouse development. *PLoS one* **6**, e18310 (2011).
4. Hu J, *et al.* Novel importin-alpha family member Kpna7 is required for normal fertility and fecundity in the mouse. *The Journal of biological chemistry* **285**, 33113-33122 (2010).
5. Gao YW, *et al.* Protein Expression Landscape of Mouse Embryos during Pre-implantation Development. *Cell reports* **21**, 3957-3969 (2017).
6. Lee EB, Lee VM, Trojanowski JQ. Gains or losses: molecular mechanisms of TDP43-mediated neurodegeneration. *Nature reviews Neuroscience* **13**, 38-50 (2011).
7. Khalil B, *et al.* Nuclear import receptors are recruited by FG-nucleoporins to rescue hallmarks of TDP-43 proteinopathy. *Molecular neurodegeneration* **17**, 80 (2022).
8. Gallardo T, Shirley L, John GB, Castrillon DH. Generation of a germ cell-specific mouse transgenic Cre line, Vasa-Cre. *Genesis (New York, NY : 2000)* **45**, 413-417 (2007).
9. Meneses A, Koga S, O'Leary J, Dickson DW, Bu G, Zhao N. TDP-43 Pathology in Alzheimer's Disease. *Molecular neurodegeneration* **16**, 84 (2021).
10. Lalmansingh AS, Urekar CJ, Reddi PP. TDP-43 is a transcriptional repressor: the testis-specific mouse *acr1* gene is a TDP-43 target in vivo. *The Journal of biological chemistry* **286**, 10970-10982 (2011).
11. Liu B, *et al.* The landscape of RNA Pol II binding reveals a stepwise transition during ZGA. *Nature* **587**, 139-144 (2020).
12. Xue J, Lou X, Ning D, Shao R, Chen G. Mechanism and treatment of α -amanitin poisoning. *Archives of toxicology* **97**, 121-131 (2023).
13. Zhang B, *et al.* Allelic reprogramming of the histone modification H3K4me3 in early mammalian development. *Nature* **537**, 553-557 (2016).

14. Kraemer BC, *et al.* Loss of murine TDP-43 disrupts motor function and plays an essential role in embryogenesis. *Acta neuropathologica* **119**, 409-419 (2010).
15. Sephton CF, *et al.* TDP-43 is a developmentally regulated protein essential for early embryonic development. *The Journal of biological chemistry* **285**, 6826-6834 (2010).
16. Wu LS, Cheng WC, Hou SC, Yan YT, Jiang ST, Shen CK. TDP-43, a neuro-pathosignature factor, is essential for early mouse embryogenesis. *Genesis (New York, NY : 2000)* **48**, 56-62 (2010).
17. Xiong Z, *et al.* Ultrasensitive Ribo-seq reveals translational landscapes during mammalian oocyte-to-embryo transition and pre-implantation development. *Nature cell biology* **24**, 968-980 (2022).
18. Oqani RK, Kim HR, Diao YF, Park CS, Jin DI. The CDK9/cyclin T1 subunits of P-TEFb in mouse oocytes and preimplantation embryos: a possible role in embryonic genome activation. *BMC developmental biology* **11**, 33 (2011).
19. Zhang C, Wang M, Li Y, Zhang Y. Profiling and functional characterization of maternal mRNA translation during mouse maternal-to-zygotic transition. *Science advances* **8**, eabj3967 (2022).

REVIEWERS' COMMENTS

Reviewer #1 (Remarks to the Author):

The authors have carried out a very thorough and detailed revision of the manuscript, and have addressed in full all of the comments previously raised. They have greatly improved the manuscript and no mayor issues remain.

Only some minor issues could be addressed before final acceptance:

1. In a couple of occasions in the Discussion, the authors mention "data not shown", referring to points raised during the revision. As these have been properly addressed and the data is made available as figure for the revisers, it might be a good idea to include this data as supplementary figures. It is true that there is already quite a lot of supplementary figures right now, so this is only a suggestion.

2. In Figure S1A, it should be RPF, not RFP.

3. There are still quite some few mistakes and typos that the authors should correct. For example, and only in the final part of the Discussion:

- "contact" instead of "contacts", line 377
- "deserve" instead of "deserved", line 386
- missing "as", line 389
- "co-localizes" in stead of "co-occupy", line 391
- "supporting" instead of "suggest", line 400
- "deserves" instead of "deserved", line 402

Reviewer #2 (Remarks to the Author):

The authors address all of my questions in detail. They performed experiments on some of my points even when I suggested just a discussion (e.g. Fig 8e). I recommend the manuscript for immediate publication. Congratulations to the authors on the great study! Keep up the good work!

We sincerely appreciate the editor and reviewers for their valuable comments which has significantly improved the quality of our manuscript. In this revised submission, we have addressed all of the issues raised in the previous review process.

For the reviewers' and editors' convenience, we also included a version of the manuscript in which the key revised sections are highlighted. The POINT-BY-POINT RESPONSES to the reviewers' comments are as follows in bold font.

POINT-BY-POINT RESPONSES

Reviewer #1:

The authors have carried out a very thorough and detailed revision of the manuscript, and have addressed in full all of the comments previously raised. They have greatly improved the manuscript and no mayor issues remain.

We greatly appreciate the reviewer's recognition of the strengths in our revised work and the constructive comments to further enhance the quality of our manuscript.

Only some minor issues could be addressed before final acceptance:

Comment 1: In a couple of occasions in the Discussion, the authors mention "data not shown", referring to points raised during the revision. As these have been properly addressed and the data is made available as figure for the revisers, it might be a good idea to include this data as supplementary figures. It is true that there is already quite a lot of supplementary figures right now, so this is only a suggestion.

RESPONSES TO COMMENT 1:

We thank the reviewer for the suggestions, we have added the Reviewer only Figure IV b,c as the Supplementary Figure 8 g,h in the revised manuscript. The data of TDP-43 in ESCs is too preliminary to published and we will further explore the role and mechanism of TDP43 in ESCs in our lab. Thus, we do not add these the data of ESCs in the revised manuscript. According to the guidance of Nature Communications, we have delete the statement "Similar phenomena were observed in mouse ESCs (data not show)" in the revised manuscript.

Comment 2: In Figure S1A, it should be RPF, not RFP.

RESPONSES TO COMMENT 2:

Thank you for your correction. We have changed RFP into RPF in Figure S1a in the revised manuscript.

Comment 3: There are still quite some few mistakes and typos that the authors should correct. For example, and only in the final part of the Discussion:

- “contact” instead of “contacts”, line 377
- “deserve” instead of “deserved”, line 386
- missing “as”, line 389
- “co-localizes” in stead of “co-occupy”, line 391
- “supporting” instead of “suggest”, line 400
- “deserves” instead of “deserved”, line 402

RESPONSES TO COMMENT 3:

We appreciate the reviewer for pointing these mistakes out. As suggested, we have correct these mistakes. And we have carefully edited English language of the revised manuscript.

Reviewer #2 (Remarks to the Author):

The authors address all of my questions in detail. They performed experiments on some of my points even when I suggested just a discussion (e.g. Fig 8e). I recommend the manuscript for immediate publication. Congratulations to the authors on the great study! Keep up the good work!

RESPONSES TO COMMENT:

We thank the reviewer very much for appreciating the merits of this work. You really gave great encouragement to us during the review. We sincerely appreciate you for your providing constructive comments to strengthen our manuscript again.